# Stability of Stochastic Gradient Descent on Nonsmooth Convex Losses

**Raef Bassily**
Department of Computer Science & Engineering
The Ohio State University.
bassily.1@osu.edu

**Vitaly Feldman**
Apple*

**Cristóbal Guzmán**
Pontificia Universidad Católica de Chile
Institute for Mathematical and Computational Engineering
ANID – Millennium Science Initiative Program
Millennium Nucleus Center for the Discovery of
Structures in Complex Data
crguzmanp@mat.uc.cl

**Kunal Talwar**
Apple*
ktalwar@apple.com

## Abstract

Uniform stability is a notion of algorithmic stability that bounds the worst case change in the model output by the algorithm when a single data point in the dataset is replaced. An influential work of Hardt et al. [20] provides strong upper bounds on the uniform stability of the stochastic gradient descent (SGD) algorithm on sufficiently smooth convex losses. These results led to important progress in understanding of the generalization properties of SGD and several applications to differentially private convex optimization for smooth losses.

Our work is the first to address uniform stability of SGD on *nonsmooth* convex losses. Specifically, we provide sharp upper and lower bounds for several forms of SGD and full-batch GD on arbitrary Lipschitz nonsmooth convex losses. Our lower bounds show that, in the nonsmooth case, (S)GD can be inherently less stable than in the smooth case. On the other hand, our upper bounds show that (S)GD is sufficiently stable for deriving new and useful bounds on generalization error. Most notably, we obtain the first dimension-independent generalization bounds for multi-pass SGD in the nonsmooth case. In addition, our bounds allow us to derive a new algorithm for differentially private nonsmooth stochastic convex optimization with optimal excess population risk. Our algorithm is simpler and more efficient than the best known algorithm for the nonsmooth case [16].

## 1 Introduction

Successful applications of a machine learning algorithm require the algorithm to generalize well to unseen data. Thus understanding and bounding the generalization error of machine learning algorithms is an area of intense theoretical interest and practical importance. The single most popular approach to modern machine learning relies on the use of continuous optimization techniques to optimize the appropriate loss function, most notably the stochastic (sub)gradient descent (SGD) method. Yet the generalization properties of SGD are still not well understood.

Consider the setting of stochastic convex optimization (SCO). In this problem, we are interested in the minimization of the population risk $F_\mathcal{D}(x) := \mathbb{E}_{\mathbf{z} \sim \mathcal{D}}[f(x, \mathbf{z})]$, where $\mathcal{D}$ is an arbitrary and unknown distribution, for which we have access to an i.i.d. sample of size $n$, $\mathbf{S} = (\mathbf{z}_1, \ldots, \mathbf{z}_n)$; and $f(\cdot, z)$ is convex and Lipschitz for all $z$. The performance of an algorithm $\mathcal{A}$ is quantified by its expected *excess population risk*,

$$\mathbb{E}[\varepsilon_{\mathrm{risk}}(\mathcal{A})] := \mathbb{E}[F_\mathcal{D}(\mathcal{A}(\mathbf{S}))] - \min_{x \in \mathcal{X}} F_\mathcal{D}(x),$$

where the expectation is taken with respect to the randomness of the sample $\mathbf{S}$ and internal randomness of $\mathcal{A}$. A standard way to bound the excess risk is given by its decomposition into optimization error (a.k.a. training error) and generalization error (see eqn. (2) in Sec. 2). The optimization error can be easily measured empirically but assessing the generalization error requires access to fresh samples from the same distribution. Thus bounds on the generalization error lead directly to provable guarantees on the excess population risk.

Classical analysis of SGD allows obtaining bounds on the excess population risk of one pass SGD. In particular, with an appropriately chosen step size, SGD gives a solution with expected excess population risk of $O(1/\sqrt{n})$ and this rate is optimal [30]. However, this analysis does not apply to multi-pass SGD that is ubiquitous in practice.

In an influential work, Hardt et al. [20] gave the first bounds on the generalization error of general forms of SGD (such as those that make multiple passes over the data). Their analysis relies on algorithmic stability, a classical tool for proving bounds on the generalization error. Specifically, they gave strong bound on the *uniform stability* of several variants of SGD on convex and smooth losses (with $2/\eta$-smoothness sufficing when all the step sizes are at most $\eta$). Uniform stability bounds the worst case change in loss of the model output by the algorithm on the worst case point when a single data point in the dataset is replaced [4]. Formally, for a randomized algorithm $\mathcal{A}$, loss functions $f(\cdot, z)$ and $S \simeq S'$ and $z \in \mathcal{Z}$, let $\gamma_\mathcal{A}(S, S', z) := f(\mathcal{A}(S), z) - f(\mathcal{A}(S'), z)$, where $S \simeq S'$ denotes that the two datasets differ only in a single data point. We say $\mathcal{A}$ is $\gamma$-uniformly stable if

$$\sup_{S \simeq S', z} \mathbb{E}[\gamma_\mathcal{A}(S, S', z)] \leq \gamma,$$

where the expectation is over the internal randomness of $\mathcal{A}$. Stronger notions of stability can also be considered, e.g., bounding the probability – over the internal randomness of $\mathcal{A}$ – that $\gamma_\mathcal{A}(S, S', z) > \gamma$. Using stability, [20] showed that several variants of SGD simultaneously achieve the optimal tradeoff between the excess empirical risk and stability with both being $O(1/\sqrt{n})$. Several works have used this approach to derive new generalization properties of SGD [28, 9, 19].

The key insight of Hardt et al. [20] is that a gradient step on a sufficiently smooth convex function is a nonexpansive operator (that is, it does not increase the $\ell_2$ distance between points). Unfortunately, this property does not hold for nonsmooth losses such as the hinge loss. As a result, no non-trivial bounds on the uniform stability of SGD have been previously known in this case.

Uniform stability is also closely related to the notion of differential privacy (DP). DP upper bounds the worst case change in the output distribution of an algorithm when a single data point in the dataset is replaced [14]. This connection has been exploited in the design of several DP algorithms for SCO. In particular, bounds on the uniform stability of SGD from [20] have been crucial in the design and analysis of new DP-SCO algorithms [36, 12, 17, 2, 16].

## 1.1 Our Results

We establish tight bounds on the uniform stability of the (stochastic) subgradient descent method on nonsmooth convex losses. These results demonstrate that in the nonsmooth case SGD can be substantially less stable. At the same time we show that SGD has strong stability properties even in the regime when its iterations can be expansive.

For convenience, we describe our results in terms of *uniform argument stability* (UAS), which bounds the output sensitivity in $\ell_2$-norm w.r.t. an arbitrary change in a single data point. Formally, a (randomized) algorithm has $\delta$-UAS if

$$\sup_{S \simeq S'} \mathbb{E} \|\mathcal{A}(S) - \mathcal{A}(S')\|_2 \leq \delta. \tag{1}$$

This notion is implicit in existing analyses of uniform stability [4, 32, 20] and was explicitly defined by Liu et al. [27]. In this work, we prove stronger – high probability – upper bounds on the random

variable $\delta_{\mathcal{A}}(S, S') := \|\mathcal{A}(S) - \mathcal{A}(S')\|$,[2] and we provide matching lower bounds for the weaker – in expectation – notion of UAS (1). A summary of our bounds is in Table 1. For simplicity, they are provided for constant step size; general step sizes (for upper bounds) are provided in Section 3.

| Algorithm | H.p. upper bound | Expectation upper bound | Expectation Lower bound |
|---|---|---|---|
| GD (full batch) | $4\big(\eta\sqrt{T} + \frac{\eta T}{n}\big)$ | $4\big(\eta\sqrt{T} + \frac{\eta T}{n}\big)$ | $\Omega\big(\eta\sqrt{T} + \frac{\eta T}{n}\big)$ |
| SGD (w/replacement) | $4\big(\eta\sqrt{T} + \frac{\eta T}{n}\big)$ | $\min\{1, \frac{T}{n}\}3\eta\sqrt{T} + 4\frac{\eta T}{n}$ | $\Omega\Big(\min\{1, \frac{T}{n}\}\eta\sqrt{T} + \frac{\eta T}{n}\Big)$ |
| SGD (fixed permutation) | $2\eta\sqrt{T} + 4\frac{\eta T}{n}$ | $\min\{1, \frac{T}{n}\}2\eta\sqrt{T} + 4\frac{\eta T}{n}$ | $\Omega\Big(\min\{1, \frac{T}{n}\}\eta\sqrt{T} + \frac{\eta T}{n}\Big)$ |

Table 1: UAS for GD and SGD. Here $T = \#$ iterations; $\eta$ is the step size.

Compared to the smooth case [20], the main difference is the presence of the additional $\eta\sqrt{T}$ term. This term has important implications for the generalization bounds derived from UAS. The first one is that the standard step size $\eta = \Theta(1/\sqrt{n})$ used in single pass SGD leads to a vacuous stability bound. Unfortunately, as shown by our lower bounds, this is unavoidable (at least in high dimension). However, by decreasing the step size and increasing the number of steps, one obtains a variant of SGD with nearly optimal balance between the UAS and the excess empirical risk.

We highlight two major consequences of our bounds:

- **Generalization bounds for multi-pass nonsmooth SGD.** We prove that the generalization error of multi-pass SGD with $K$ passes is bounded by $O((\sqrt{Kn} + K)\eta)$. This result can be easily combined with training error guarantees to provide excess risk bounds for this algorithm. Since training error can be measured directly, our generalization bounds would immediately yield strong guarantees on the excess risk in practical scenarios where we can certify small training error.
- **Differentially private stochastic convex optimization for non-smooth losses.** We show that a variant of standard noisy SGD [3] with constant step size and $n^2$ iterations yields the optimal excess population risk $O\big(\frac{1}{\sqrt{n}} + \frac{\sqrt{d\log(1/\beta)}}{\alpha n}\big)$ for convex *nonsmooth* losses under $(\alpha, \beta)$-differential privacy. The best previous algorithm for this problem is substantially more involved: it relies on a multi-phase regularized SGD with decreasing step sizes and variable noise rates and uses $O(n^2\sqrt{\log(1/\beta)})$ gradient computations [16].

## 1.2 Overview of Techniques

- **Upper bounds.** When gradient steps are nonexpansive, upper-bounding UAS requires simply summing the differences between the gradients on the neighboring datasets when the replaced data point is used [20]. This gives the bound of $\eta T/n$ in the smooth case.

  By contrast, in the nonsmooth case, UAS may increase even when the gradient step is performed on the same function. As a result it may increase in *every single iteration*. However, we use the fact that the difference in the subgradients has negative inner product with the difference between the iterates themselves (by monotonicity of the subgradient). Thus the increase in distance satisfies a recurrence with a quadratic and a linear term. Solving this recurrence leads to our upper bounds.
- **Lower bounds.** The lower bounds are based on a function with a highly nonsmooth behavior around the origin. More precisely, it is the maximum of linear functions plus a small linear drift that is controlled by a single data point. We show that, when starting the algorithm from the origin, the presence of the linear drift pushes the iterate into a trajectory in which each subgradient step is orthogonal to the current iterate. Thus, if $d \geq \min\{T, 1/\eta^2\}$, we get the $\sqrt{T}\eta$ increase in UAS. Our lower bounds are also robust to averaging of the iterates. The detailed constructions and analyses can be found in the full version of the paper.

## 1.3 Other Related Work

Stability is a classical approach to proving generalization bounds pioneered by Rogers and Wagner [31] and Devroye and Wagner [10, 11]. It is based on analysis of the sensitivity of the learning

algorithm to changes in the dataset such as leaving one of the data points out or replacing it with a different one. Uniform stability was introduced by Bousquet and Elisseeff [4] in order to derive general bounds on the generalization error that hold with high probability. These bounds have been significantly improved in a recent sequence of works [18, 19, 5]. Chen et al. [9] establish limits of stability in the smooth convex setting, proving that accelerated methods must satisfy strong stability lower bounds. Stability-based data-dependent generalization bounds for continuous losses were studied in [29, 25]. Independently and in concurrent work, Lei et al. [26] study *on-average* argument stability of SGD with Hölder continuous gradients and some nonconvex settings. Their *on-average* stability bounds match our uniform stability bounds in the nonsmooth convex case. Note however that on-average stability is a weaker notion of stability than ours. In [26, Appendix G], a high probability stability bound is derived for particular vanishing stepsize sequences; however, they lead to constant excess risk bounds in the nonsmooth convex case (and for constant stepsize they incur in an additional $\sqrt{T}$ factor, compared to our bounds). In comparison, our bounds are more general (for arbitrary stepsizes); moreover, we prove the optimality of our bounds up to constant factors.

First applications of uniform stability in the context of stochastic convex optimization relied on the stability of the empirical minimizer for strongly convex losses [4]. Therefore a natural approach to achieve uniform stability (and also UAS) is to add a strongly convex regularizer and solve the ERM to high accuracy [32]. Recent applications of this approach can be found for example in [24, 6, 16]. In contrast, our approach does not require strong convexity and applies to all iterates of the SGD and not only to a very accurate empirical minimizer.

Classical approach to generalization relies on *uniform convergence* of empirical risk to population risk. Unfortunately, without additional structural assumptions on convex functions, a lower bound of $\Omega(\sqrt{d/n})$ on the rate of *uniform convergence* for convex SCO is known [32, 15]. The dependence on the dimension $d$ makes the bound obtained via the uniform convergence approach vacuous in the high-dimensional settings common in modern applications.

Differentially private convex optimization has been studied extensively for over a decade (see, e.g., [7, 8, 21, 23, 33, 3, 35, 22, 34, 2, 16]). However, until recently, the research focused on minimization of the empirical risk. A recent work of Bassily et al. [2] established that the optimal rate of the excess population risk for $(\alpha, \beta)$-DP SCO algorithms is $O\left(\frac{1}{\sqrt{n}} + \frac{\sqrt{d \log(1/\beta)}}{\alpha n}\right)$. Their algorithms are relatively inefficient, especially in the nonsmooth case. Subsequently, Feldman et al. [16] gave several new algorithms for DP-SCO with the optimal population risk. For sufficiently smooth losses, their algorithms use a linear number of gradient computations. In the nonsmooth case, as mentioned earlier, their algorithm requires $O(n^2 \sqrt{\log(1/\beta)})$ gradient computations and is significantly more involved than the algorithm shown here.

## 2  Notation and Preliminaries

Throughout we work on the Euclidean space $(\mathbb{R}^d, \|\cdot\|_2)$. We denote the Euclidean ball of radius $r > 0$ centered at $x \in \mathbb{R}^d$ by $\mathcal{B}(x, r)$. In what follows, we assume $\mathcal{X} \subseteq \mathcal{B}(0, R)$ is a closed, convex set for some $R > 0$. Let $\mathsf{Proj}_{\mathcal{X}}$ be the Euclidean projection onto $\mathcal{X}$. A convex function $f : \mathcal{X} \mapsto \mathbb{R}$ is $L$-Lipschitz if

$$f(x) - f(y) \le L\|x - y\| \qquad (\forall x, y \in \mathcal{X}).$$

We denote the class of convex $L$-Lipschitz functions as $\mathcal{F}_{\mathcal{X}}^0(L)$. In this work, we will focus on the class $\mathcal{F}_{\mathcal{X}}^0(L)$ defined over a compact convex set $\mathcal{X}$. Since the Euclidean radius of $\mathcal{X}$ is bounded by $R$, we will assume that the range of these functions lies in $[-RL, RL]$.

**Nonsmooth stochastic convex optimization:** We study the standard setting of nonsmooth stochastic convex optimization

$$x^* \in \arg\min\{F_{\mathcal{D}}(x) := \mathbb{E}_{\mathbf{z} \sim \mathcal{D}}[f(x, \mathbf{z})] : \ x \in \mathcal{X}\}.$$

Here, $\mathcal{D}$ is an unknown distribution supported on a set $\mathcal{Z}$, and $f(\cdot, z) \in \mathcal{F}_{\mathcal{X}}^0(L)$ for all $z \in \mathcal{Z}$. In the stochastic setting, we assume access to an i.i.d. sample from $\mathcal{D}$, denoted as $\mathbf{S} = (\mathbf{z}_1, \ldots, \mathbf{z}_n) \sim \mathcal{D}^n$.

**A stochastic optimization algorithm** is a (randomized) mapping $\mathcal{A} : \mathcal{Z}^n \mapsto \mathcal{X}$. When the algorithm is randomized, $\mathcal{A}(\mathbf{S})$ is a random variable depending on both the sample $\mathbf{S} \sim \mathcal{D}^n$ and its own random coins. The performance of $\mathcal{A}$ is quantified by its *excess population risk*

$$\varepsilon_{\mathrm{risk}}(\mathcal{A}) := F_{\mathcal{D}}(\mathcal{A}(\mathbf{S})) - F_{\mathcal{D}}(x^*).$$

Note that $\varepsilon_{\text{risk}}(\mathcal{A})$ is a random variable (due to randomness in the sample $\mathbf{S}$ and any possible internal randomness of the algorithm).

**Empirical risk minimization (ERM)** is one of the most standard approaches to stochastic convex optimization. In the ERM problem, we are given a sample $\mathbf{S} = (\mathbf{z}_1, \ldots, \mathbf{z}_n)$, and the goal is to find

$$x^*(\mathbf{S}) \in \arg\min \left\{ F_{\mathbf{S}}(x) := \frac{1}{n} \sum_{i=1}^{n} f(x, \mathbf{z}_i) : \ x \in \mathcal{X} \right\}.$$

**Risk decomposition:** A common way to bound the excess population risk is by decomposing it into *generalization*, *optimization* and *approximation error*:

$$\varepsilon_{\text{risk}}(\mathcal{A}) \leq \underbrace{F_{\mathcal{D}}(\mathcal{A}(\mathbf{S})) - F_{\mathbf{S}}(\mathcal{A}(\mathbf{S}))}_{\varepsilon_{\text{gen}}(\mathcal{A})} + \underbrace{F_{\mathbf{S}}(\mathcal{A}(\mathbf{S})) - F_{\mathbf{S}}(x^*(\mathbf{S}))}_{\varepsilon_{\text{opt}}(\mathcal{A})} + \underbrace{F_{\mathbf{S}}(x^*(\mathbf{S})) - F_{\mathcal{D}}(x^*)}_{\varepsilon_{\text{approx}}}. \quad (2)$$

Here, the optimization error corresponds to the empirical optimization gap, which can be bounded by standard optimization convergence analysis. The expected value of the approximation error is at most zero. One can show, e.g., by Hoeffding's inequality, that the approximation error is bounded by $\tilde{O}(LR/\sqrt{n})$ with high probability (see full version). Therefore, to establish bounds on the excess risk it suffices to upper bound the optimization and generalization errors.

We say that two datasets $S, \ S'$ are neighboring, denoted $S \simeq S'$, if they only differ on a single entry; i.e., there exists $i \in [n]$ s.t. for all $k \neq i$, $z_k = z_k'$.

**Uniform argument stability (UAS):** Given an algorithm $\mathcal{A}$ and datasets $S \simeq S'$, we define the *uniform argument stability* (UAS) random variable as

$$\delta_{\mathcal{A}}(S, S') := \|\mathcal{A}(S) - \mathcal{A}(S')\|.$$

The randomness here is due to any possible internal randomness of $\mathcal{A}$. For any $L$-Lipschitz function $f$, we have that $f(\mathcal{A}(S), z) - f(\mathcal{A}(S'), z) \leq L \, \delta_{\mathcal{A}}(S, S')$. Hence, upper bounds on UAS can be easily transformed into upper bounds on uniform stability.

In this work, we will consider two types of bounds on UAS.

**High-probability guarantees on UAS:** For any pair $S \simeq S'$, one can bound the UAS random variable $\delta_{\mathcal{A}}(S, S')$ w.h.p. over the internal randomness of $\mathcal{A}$. High-probability upper bounds on UAS lead to high-probability upper bounds on generalization error $\varepsilon_{\text{gen}}$. We will use the following theorem, which follows in a straightforward fashion from [19, Theorem 1.1], to derive generalization-error guarantees for our results in Sections 4 and 5 based on our UAS upper bounds in Section 3.

**Theorem 2.1** (follows from Theorem 1.1 in [19]). *Let $\mathcal{A} : \mathcal{Z}^n \to \mathcal{X}$ be a randomized algorithm. Suppose that the UAS random variable of $\mathcal{A}$ satisfies:*

$$\mathbb{P}_{\mathcal{A}} \left[ \sup_{S \simeq S'} \delta_{\mathcal{A}}(S, S') \geq \gamma \right] \leq \theta_0.$$

*Then there is a constant $c$ such that for any distribution $\mathcal{D}$ over $\mathcal{Z}$ and any $\theta \in (0, 1)$, we have*

$$\mathbb{P}_{\mathbf{S} \sim \mathcal{D}^n, \, \mathcal{A}} \left[ |\varepsilon_{\text{gen}}(\mathcal{A})| \geq c \left( L\gamma \log(n) \log(n/\theta) + LR\sqrt{\frac{\log(1/\theta)}{n}} \right) \right] \leq \theta + \theta_0,$$

*where $\varepsilon_{\text{gen}}(\mathcal{A}) = F_{\mathcal{D}}(\mathcal{A}(\mathbf{S})) - F_{\mathbf{S}}(\mathcal{A}(\mathbf{S}))$ as defined earlier.*

**Expectation guarantees on UAS:** Our results also include upper and lower bounds on $\sup_{S \simeq S'} \mathbb{E}_{\mathcal{A}} [\delta_{\mathcal{A}}(S, S')]$; that is the supremum of the expected value of the UAS random variable, where the supremum is taken over all pairs of neighboring datasets. This suffices to obtain generalization error guarantees in expectation [20].

## 3 Tight Bounds on Uniform Argument Stability

In this section we establish sharp bounds on the Uniform Argument Stability for SGD with nonsmooth convex losses. We start our analysis by a key lemma, which can be used to bound the UAS for generic versions of SGD. Next we present one of our main results that establishes the first sharp bound on the UAS of nonsmooth SGD. We only provide details for the sampling-with-replacement SGD method, but full-batch GD and fixed-permutation SGD are analyzed similarly in the full version.

## 3.1 The Basic Lemma

We begin by stating a key lemma that encompasses the UAS bound analysis of multiple variants of (S)GD. In particular, all of our UAS upper bounds are obtained by almost a direct application of this lemma. In the lemma we consider two gradient descent trajectories associated to different sequences of objective functions. The degree of concordance of the two sequences, quantified by the distance between the subgradients at the current iterate, controls the deviation between the trajectories. We note that this distance condition is satisfied for all (S)GD variants we study in this work.

**Lemma 3.1.** *Let* $(x^t)_{t\in[T]}$ *and* $(y^t)_{t\in[T]}$, *with* $x^1 = y^1$, *be online gradient descent trajectories for convex $L$-Lipschitz objectives* $(f_t)_{t\in[T-1]}$ *and* $(f'_t)_{t\in[T-1]}$, *respectively; i.e.,*

$$
\begin{aligned}
x^{t+1} &= \mathsf{Proj}_{\mathcal{X}}[x^t - \eta_t \nabla f_t(x^t)]\\
y^{t+1} &= \mathsf{Proj}_{\mathcal{X}}[y^t - \eta_t \nabla f'_t(y^t)],
\end{aligned}
$$

*for all* $t \in [T-1]$. *Suppose for every* $t \in [T-1]$, $\|\nabla f_t(x^t) - \nabla f'_t(x^t)\| \le a_t$, *for scalars* $0 \le a_t \le 2L$. *Then, if* $t_0 = \inf\{t : f_t \ne f'_t\}$,

$$
\|x^T - y^T\| \le 2L\sqrt{\sum_{t=t_0}^{T-1} \eta_t^2} + 2\sum_{t=t_0+1}^{T-1} \eta_t a_t.
$$

*Proof.* Let $\delta_t = \|x^t - y^t\|$. By definition of $t_0$ it is clear that $\delta_1 = \ldots = \delta_{t_0} = 0$. For $t = t_0 + 1$, we have that $\delta_{t_0+1} = \|\eta_{t_0}(\nabla f_{t_0}(x^{t_0}) - \nabla f'_{t_0}(y^{t_0}))\| \le 2L\eta_{t_0}$.

Now, we derive a recurrence for $(\delta_t)_{t\in[T]}$:

$$
\begin{aligned}
\delta_{t+1}^2 = \|\mathsf{Proj}_{\mathcal{X}}[x^t - \eta_t \nabla f_t(x^t)] - \mathsf{Proj}_{\mathcal{X}}[y^t - \eta_t \nabla f'_t(y^t)]\|^2 &\le \|x^t - y^t - \eta_t(\nabla f_t(x^t) - \nabla f'_t(y^t))\|^2\\
= \delta_t^2 + \eta_t^2\|\nabla f_t(x^t) - \nabla f'_t(y^t)\|^2 &- 2\eta_t\langle \nabla f_t(x^t) - \nabla f'_t(y^t), x^t - y^t\rangle\\
\le \delta_t^2 + \eta_t^2\|\nabla f_t(x^t) - \nabla f'_t(y^t)\|^2 &- 2\eta_t\langle \nabla f_t(x^t) - \nabla f'_t(x^t), x^t - y^t\rangle - 2\eta_t\langle \nabla f'_t(x^t) - \nabla f'_t(y^t), x^t - y^t\rangle\\
\le \delta_t^2 + \eta_t^2\|\nabla f_t(x^t) - \nabla f'_t(y^t)\|^2 &+ 2\eta_t\|\nabla f_t(x^t) - \nabla f'_t(x^t)\|\delta_t - 2\eta_t\langle \nabla f'_t(x^t) - \nabla f'_t(y^t), x^t - y^t\rangle\\
\le \delta_t^2 + 4L^2\eta_t^2 + 2\eta_t a_t \delta_t,
\end{aligned}
$$

where at the last step we use the monotonicity of the subgradient. Note that

$$
\delta_{t_0+1} \le \eta_{t_0}\|\nabla f_{t_0}(x^{t_0}) - \nabla f'_{t_0}(x^{t_0})\| \le 2L\eta_{t_0}.
$$

Hence,

$$
\begin{aligned}
\delta_t^2 &\le \delta_{t_0+1}^2 + 4L^2\sum_{s=t_0+1}^{t-1}\eta_s^2 + 2\sum_{s=t_0+1}^{t-1}\eta_s a_s \delta_s\\
&\le 4L^2\sum_{s=t_0}^{t-1}\eta_s^2 + 2\sum_{s=t_0+1}^{t-1}\eta_s a_s \delta_s.
\end{aligned} \tag{3}
$$

Now we prove the following bound by induction (notice this claim proves the result):

$$
\delta_t \le 2L\sqrt{\sum_{s=t_0}^{t-1}\eta_s^2} + 2\sum_{s=t_0+1}^{t-1}\eta_s a_s \qquad (\forall t \in [T]).
$$

Indeed, the claim is clearly true for $t = t_0$. For the inductive step, we assume it holds for some $t \in [T-1]$. To prove the result we consider two cases: first, when $\delta_{t+1} \le \max_{s\in[t]}\delta_s$, by induction hypothesis we have

$$
\delta_{t+1} \le \delta_t \le 2L\sqrt{\sum_{s=t_0}^{t-1}\eta_s^2} + 2\sum_{s=t_0+1}^{t-1}\eta_s a_s \le 2L\sqrt{\sum_{s=t_0}^{t}\eta_s^2} + 2\sum_{s=t_0+1}^{t}\eta_s a_s.
$$

In the other case, $\delta_{t+1} > \max_{s\in[t]}\delta_s$, we use (3)

$$
\delta_{t+1}^2 \le 4L^2\sum_{s=t_0}^{t}\eta_t^2 + 2\sum_{s=t_0+1}^{t}\eta_s a_s \delta_s \le 4L^2\sum_{s=t_0}^{t}\eta_t^2 + 2\delta_{t+1}\sum_{s=t_0+1}^{t}\eta_s a_s,
$$

which is equivalent to

$$
\left(\delta_{t+1} - \sum_{s=t_0+1}^{t}a_s\eta_s\right)^2 \le 4L^2\sum_{s=t_0}^{t}\eta_t^2 + \left(\sum_{s=t_0+1}^{t}\eta_s a_s\right)^2.
$$

Taking square root at this inequality, and using the subadditivity of the square root, we obtain the inductive step, and therefore the result. □

Theorem 3.2 below summarizes our results for the UAS of sampling-with-replacement SGD on nonsmooth losses. Given our key lemma, we prove both in-expectation and high-probability upper bounds on the UAS. Our theorem below also establishes the tightness of our upper bounds by showing a matching lower bound. Detailed statements of these results and their full proofs are deferred to the full version of the paper. Similar analysis of full-batch GD and fixed-permutation SGD are also deferred to the full version of the paper.

---

**Algorithm 1** $\mathcal{A}_{\mathsf{rSGD}}$: Sampling with replacement SGD

---

**Require:** Dataset: $S = (z_1, \ldots, z_n) \in \mathcal{Z}^n$, # iterations $T$, stepsizes $\{\eta_t : t \in [T]\}$
1: Choose arbitrary initial point $x^1 \in \mathcal{X}$
2: **for** $t = 1$ to $T - 1$ **do**
3:     Sample $\mathbf{i}_t \sim \mathsf{Unif}([n])$
4:     $x^{t+1} := \mathsf{Proj}_{\mathcal{X}} \left( x^t - \eta_t \cdot \nabla f(x^t, z_{\mathbf{i}_t}) \right)$
5: **return** $\overline{x}^T = \frac{1}{\sum_{t \in [T]} \eta_t} \sum_{t \in [T]} \eta_t x^t$

---

**Theorem 3.2** (Sharp bounds on UAS for nonsmooth SGD). *Let $\mathcal{X} \subseteq \mathcal{B}(0, R)$ and $\mathcal{F} = \mathcal{F}_{\mathcal{X}}^0(L)$. The uniform argument stability of the sampling-with-replacement SGD (Algorithm 1) satisfies:*

$$\sup_{S \simeq S'} \mathop{\mathbb{E}}_{\mathcal{A}_{\mathsf{rSGD}}} [\delta_{\mathcal{A}_{\mathsf{rSGD}}}(S, S')] \le \min \left( 2R, \; L \left( 2 \sqrt{\sum_{t=1}^{T-1} \eta_t^2} + \frac{4}{n} \sum_{t=1}^{T-1} \eta_t \right) \right).$$

*Moreover, if $\eta_t = \eta > 0 \; \forall t$ then, with probability at least $1 - n \exp(-n/2)$ (over the algorithm's internal randomness), the UAS random variable satisfies*

$$\sup_{S \simeq S'} \delta_{\mathcal{A}_{\mathsf{rSGD}}}(S, S') \le \min \left( 2R, \; 4L \left( \eta \sqrt{T-1} + \eta \frac{T-1}{n} \right) \right).$$

*Finally, if $d \ge \min\{T, 1/\eta^2\}$ and assuming[3] $T \ge n$, there exist $S \simeq S'$ such that Algorithm 1 with constant stepsize $\eta_t = \eta > 0$ satisfies $\mathop{\mathbb{E}}_{\mathcal{A}_{\mathsf{rSGD}}} [\delta_{\mathcal{A}_{\mathsf{rSGD}}}(S, S')] = \Omega \left( \min \left( R, \; L \left( \eta \sqrt{T} + \frac{\eta T}{n} \right) \right) \right).$*

## 4 Generalization Guarantees for Multi-pass SGD

One important implication of our stability bounds is that they provide non-trivial generalization error guarantees for multi-pass SGD on nonsmooth losses. Multi-pass SGD is one of the most extensively used settings of SGD in practice, where SGD is run for $K$ passes (epochs) over the dataset (namely, the number of iterations $T = Kn$). To the best of our knowledge, aside from the dimension-dependent bounds based on uniform convergence [32], no generalization error guarantees are known for the multi-pass setting on general nonsmooth convex losses. Given our uniform stability upper bounds, we can prove the following generalization error guarantees for the multi-pass setting of sampling-with-replacement SGD. Analogous results can be obtained for fixed-permutation SGD (using our stability bounds for the latter in Sec. 3 of the full version).

**Theorem 4.1.** *Running Algorithm 1 for $K$ passes (i.e., for $T = Kn$ iterations) with constant stepsize $\eta_t = \eta > 0$ yields the following generalization error guarantees:*

$$|\mathbb{E}_{\mathcal{A}_{\mathsf{rSGD}}}[\varepsilon_{\mathrm{gen}}(\mathcal{A}_{\mathsf{rSGD}})]| \le 4L^2 \eta \left( \sqrt{Kn} + K \right),$$

*and there exists $c > 0$, such that for any $0 < \theta < 1$, with probability $\ge 1 - \theta - n \exp(-n/2)$,*

$$|\varepsilon_{\mathrm{gen}}(\mathcal{A}_{\mathsf{rSGD}})| \le c \left( L^2 \eta \left( \sqrt{Kn} + K \right) \log(n) \log(n/\theta) + LR \sqrt{\frac{\log(1/\theta)}{n}} \right).$$

*Proof.* First, by the expectation guarantee on UAS given in Theorem 3.2 together with the fact that the losses are $L$-Lipschitz, it follows that Algorithm 1 (when run for $K$ passes with constant stepsize $\eta$) is $\gamma$-uniformly stable, where $\gamma = 4L^2 \left( \eta \sqrt{Kn} + \eta K \right)$. Then, by [20, Thm. 2.2], we have

$$|\mathbb{E}_{\mathcal{A}_{\mathsf{rSGD}}}[\varepsilon_{\mathrm{gen}}(\mathcal{A}_{\mathsf{rSGD}})]| \le \gamma.$$

For the high-probability bound, we combine the high-probability guarantee on UAS given in Theorem 3.2 with Theorem 2.1 to get the claimed bound. □

These bounds on generalization error can be used to obtain excess risk bounds using the standard risk decomposition (see (2)). In practical scenarios where one can certify small optimization error for multi-pass SGD, Thm. 4.1 can be used to readily estimate the excess risk. In the full version of the paper we provide worst-case analysis showing that multi-pass SGD is guaranteed to attain the optimal excess risk of $\approx LR/\sqrt{n}$ within $n$ passes (for appropriately chosen constant stepsize).

## 5 A Simple Algorithm for Differentially Private Nonsmooth Stochastic Convex Optimization with Optimal Risk

Now we show an application of our stability upper bound to *differentially private* stochastic convex optimization (DP-SCO). Here, the input sample to the stochastic convex optimization algorithm is a sensitive and private data set, thus the algorithm is required to satisfy the notion of $(\alpha, \beta)$-differential privacy. A randomized algorithm $\mathcal{A}$ is $(\alpha, \beta)$-differentially private if, for any pair of datasets $S \simeq S'$, and for all events $\mathcal{O}$ in the output range of $\mathcal{A}$, we have $\mathbb{P}\left[\mathcal{A}(S) \in \mathcal{O}\right] \le e^{\alpha} \cdot \mathbb{P}\left[\mathcal{A}(S') \in \mathcal{O}\right] + \beta$, where the probability is taken over the random coins of $\mathcal{A}$ [14, 13]. For meaningful privacy guarantees, the typical settings of the privacy parameters are $\alpha < 1$ and $\beta \ll 1/n$.

Using our UAS upper bounds, we show that a simple variant of noisy SGD [3], that requires only $n^2$ gradient computations, yields the optimal excess population risk for DP-SCO. In terms of running time, this is a small improvement over the algorithm of [16] for the nonsmooth case, which requires $O(n^2\sqrt{\log 1/\beta})$ gradient computations. More importantly, our algorithm is substantially simpler. For comparison, the algorithm in [16] is based on a multi-phase SGD, where in each phase a separate regularized ERM problem is solved. To ensure privacy, the output of each phase is perturbed with an appropriately chosen amount of noise before being used as the initial point for the next phase.

The description of the algorithm is given in Algorithm 2.

---
**Algorithm 2** $\mathcal{A}_{\mathsf{NSGD}}$: Noisy SGD for convex losses
---
**Require:** Private dataset $S = (z_1, \dots, z_n) \in \mathcal{Z}^n$, step size $\eta$; privacy parameters $\alpha \le 1$, $\beta \ll 1/n$

1: Set noise variance $\sigma^2 := \frac{8 L^2 \log(1/\beta)}{\alpha^2}$
2: Choose an arbitrary initial point $x^1 \in \mathcal{X}$
3: **for** $t = 1$ to $n^2 - 1$ **do**
4:     Sample $\mathbf{i}_t \sim \mathrm{Unif}([n])$
5:     $x^{t+1} := \mathsf{Proj}_{\mathcal{X}}\left(x^t - \eta \cdot (\nabla \ell(x^t, z_{\mathbf{i}_t}) + \mathbf{G}_t)\right)$, where $\mathbf{G}_t \sim \mathcal{N}\left(\mathbf{0}, \sigma^2 \mathbb{I}_d\right)$ drawn independently each iteration
6: **return** $\overline{x} = \frac{1}{n^2} \sum_{t=1}^{n^2} x^t$
---

We state the guarantees of Algorithm 2 below.

**Theorem 5.1** (Privacy guarantee of $\mathcal{A}_{\mathsf{NSGD}}$)**.** *Algorithm 2 is $(\alpha, \beta)$-differentially private.*

The proof of the theorem follows the same lines of [3, Theorem 2.1], but we replace their privacy analysis of the Gaussian mechanism with the tighter Moments Accountant method of [1]. We note that even though the algorithm of Abadi et al. [1] employs Poisson sampling rather than uniform sampling, their privacy analysis is still applicable in our case. In particular, the only place where sampling is invoked in their analysis is in [1, Lemma 3]. In the proof of that lemma, it is easy to see that the only relevant condition that involves sampling is satisfied in our case.

**Theorem 5.2** (Risk of $\mathcal{A}_{\mathsf{NSGD}}$)**.** *In Algorithm 2, let $\eta = R / \left(L \cdot n \cdot \max\left(\sqrt{n}, \frac{\sqrt{d \log(1/\beta)}}{\alpha}\right)\right)$. Then, for any $\theta \in (6n \exp(-n/2), 1)$, with probability at least $1 - \theta$ over the randomness in both the sample and the algorithm, we have*

$$\varepsilon_{\mathsf{risk}}(\mathcal{A}_{\mathsf{NSGD}}) = RL \cdot O\left(\max\left(\frac{\log(n)\log(n/\theta)}{\sqrt{n}}, \frac{\sqrt{d \log(1/\beta)}}{\alpha n}\right)\right)$$

***Proof outline:*** We first use standard online-to-batch conversion technique to provide a high-probability bound on $\varepsilon_{\text{opt}}(\mathcal{A}_{\text{NSGD}})$ (excess empirical error of $\mathcal{A}_{\text{NSGD}}$). We next observe that our high-probability upper bound on UAS in Theorem 3.2 applies directly to $\mathcal{A}_{\text{NSGD}}$ since noise addition does not impact the stability analysis. By Theorem 2.1, this implies a high-probability bound on the generalization error $\varepsilon_{\text{gen}}(\mathcal{A}_{\text{NSGD}})$. Using the standard risk decomposition (see eq. (2)), we get a bound on the excess population risk. Optimizing this bound in $\eta$ yields the claimed bound value in Theorem 5.2 (for the value of $\eta$ in the theorem statement).

Full proofs of the theorems above are deferred to the full version (see Section 6.1 therein).

**Remark 5.3.** *Using the expectation guarantee on UAS given in Theorem 3.2 and following similar steps of the analysis above, we can also show that the expected excess population risk of $\mathcal{A}_{\text{NSGD}}$ is bounded as:*

$$\mathbb{E}\left[\varepsilon_{\text{risk}}\left(\mathcal{A}_{\text{NSGD}}\right)\right] = RL \cdot O\left(\max\left(\frac{1}{\sqrt{n}}, \frac{\sqrt{d\log(1/\beta)}}{\alpha\,n}\right)\right).$$

## 6 Discussion and Open Problems

In this work we provide sharp upper and lower bounds on uniform argument stability for the (stochastic) subgradient method in stochastic nonsmooth convex optimization. Our lower bounds show inherent limitations of stability bounds compared to the smooth convex case, however we can still derive optimal population risk bounds by reducing the step size and running the algorithms for a longer number of iterations. We provide applications of this idea for differentially-private noisy SGD, and for two versions of multipass SGD (sampling-with-replacement and fixed-permutation).

The first open problem regards lower bounds that are robust to general forms of algorithmic randomization. Unfortunately, the methods presented here are not robust in this respect, since random initialization would prevent the trajectories from reaching the region of highly nonsmooth behavior of the objective (or doing it in such a way that it does not increase UAS). One may try to strengthen the lower bound by using a random rotation of the objective; however, this leads to an uninformative lower bound. Finding distributional constructions for lower bounds against randomization is a very interesting future direction.

Our privacy application provides optimal excess risk for an algorithm that runs for $n^2$ steps, which is impractical for large datasets. Other algorithms, e.g. in [16], run into similar limitations. Investigating whether quadratic running time is necessary for general nonsmooth DP-SCO is a very interesting future direction, that can be formalized in terms of the oracle complexity of stochastic convex optimization [30] under privacy constraints. We note that an excess risk of $\approx \sqrt{d/n}$ is attainable in linear time. For example, this can be achieved by running the noisy SGD algorithm of [3] for $n$ iterations with privacy parameter $\epsilon \approx 1/\sqrt{n}$, and use the algorithmic stability of differential privacy itself to bound the generalization error. This excess risk is clearly suboptimal unless the desired privacy parameter is very small.

## Broader Impact

Our work is theoretical in nature. There are no immediate ethical or societal consequences for the research presented here. We hope our results can offer theoretical insights that lead to a deeper understanding of a basic algorithm of central importance to modern machine learning. Our results can also have a direct impact on the design and analysis of differentially-private stochastic gradient methods, which are widely used for private data analysis.

## Acknowledgments and Disclosure of Funding

Part of this work was done while the authors were visiting the Simons Institute for the Theory of Computing during the "Data Privacy: Foundations and Applications" program. RB's research is supported by NSF Awards AF-1908281, SHF-1907715, and Google Faculty Research Award. Work by CG was partially funded by ANID – Millennium Science Initiative Program – NCN17_059. CG would like to thank Nicolas Flammarion and Juan Peypouquet for extremely valuable discussions at early stages of this work.

## Footnotes

*Part of this research was done while the author was at Google Research.

[2]In fact, for both GD and fixed-permutation SGD we can obtain w.p. 1 upper bounds on $\delta_{\mathcal{A}}(S, S')$, whereas for sampling-with-replacement SGD, we obtain a high-probability upper bound.

[3]The assumption that $T \ge n$ is not necessary. It is invoked here only to simplify the form of the bound (see Table 1 or the full version of the paper for the general version of the bounds.)

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
