[Supplementary Material]

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

## 2 Notation and Preliminaries

Throughout we work on the Euclidean space $(\mathbb{R}^d, \|\cdot\|_2)$. Therefore, we use unambiguously $\|\cdot\| = \|\cdot\|_2$. Vectors are denoted by lower case letters, e.g. $x, y$. Random variables (either scalar or vector) are denoted by boldface letters, e.g. $\mathbf{z}, \mathbf{u}$. We denote the Euclidean ball of radius $r > 0$ centered at $x \in \mathbb{R}^d$ by $\mathcal{B}(x, r)$. In what follows, $\mathcal{X} \subseteq \mathbb{R}^d$ is a compact convex set, and assume we know its Euclidean radius $R > 0$, $\mathcal{X} \subseteq \mathcal{B}(0, R)$. Let $\mathsf{Proj}_{\mathcal{X}}$ be the Euclidean projection onto $\mathcal{X}$, which is *nonexpansive* $\|\mathsf{Proj}_{\mathcal{X}}(x) - \mathsf{Proj}_{\mathcal{X}}(y)\| \leq \|x - y\|$. A convex function $f : \mathcal{X} \mapsto \mathbb{R}$ is $L$-Lipschitz if

$$f(x) - f(y) \leq L\|x - y\| \qquad (\forall x, y \in \mathcal{X}). \tag{2}$$

Functions with these properties are guaranteed to be subdifferentiable. Moreover, in the convex case, property (2) is "almost" equivalent to having subgradients bounded as $\partial f(x) \subseteq \mathcal{B}(0, L)$, for all $x \in \mathcal{X}$.[2] We denote the class of convex $L$-Lipschitz functions as $\mathcal{F}_{\mathcal{X}}^0(L)$. With slight abuse of notation, given a function $f \in \mathcal{F}_{\mathcal{X}}^0(L)$, we will denote by $\nabla f(x)$ an arbitrary choice of $g \in \partial f(x)$. In this work, we will focus on the class $\mathcal{F}_{\mathcal{X}}^0(L)$ defined over a compact convex set $\mathcal{X}$. Since the Euclidean radius of $\mathcal{X}$ is bounded by $R$, we will assume that the range of these functions lies in $[-RL, RL]$.

A convex and differentiable function $f : \mathcal{X} \mapsto \mathbb{R}$ is said to be $\mu$-smooth if

$$\|\nabla f(x) - \nabla f(y)\| \leq \mu\|x - y\| \qquad (\forall x, y \in \mathcal{X}),$$

and we denote the class of convex $\mu$-smooth functions by $\mathcal{F}_{\mathcal{X}}^1(\mu)$.

**Nonsmooth stochastic convex optimization:** We study the standard setting of nonsmooth stochastic convex optimization

$$x^* \in \arg\min\{F_{\mathcal{D}}(x) := \mathbb{E}_{\mathbf{z} \sim \mathcal{D}}[f(x, \mathbf{z})] : x \in \mathcal{X}\}.$$

Here, $\mathcal{D}$ is an unknown distribution supported on a set $\mathcal{Z}$, and $f(\cdot, z) \in \mathcal{F}_{\mathcal{X}}^0(L)$ for all $z \in \mathcal{Z}$. In the stochastic setting, we assume access to an i.i.d. sample from $\mathcal{D}$, denoted as $\mathbf{S} = (\mathbf{z}_1, \ldots, \mathbf{z}_n) \sim \mathcal{D}^n$. Here, we will use the bold symbol $\mathbf{S}$ to denote a random sample from the unknown distribution. A fixed (not random) dataset from $\mathcal{Z}^n$ will be denoted as $S = (z_1, \ldots, z_n) \in \mathcal{Z}^n$.

**A stochastic optimization algorithm** is a (randomized) mapping $\mathcal{A} : \mathcal{Z}^n \mapsto \mathcal{X}$. When the algorithm is randomized, $\mathcal{A}(\mathbf{S})$ is a random variable depending on both the sample $\mathbf{S} \sim \mathcal{D}^n$ and its own random coins. The performance of $\mathcal{A}$ is quantified by its *excess population risk*

$$\varepsilon_{\text{risk}}(\mathcal{A}) := F_{\mathcal{D}}(\mathcal{A}(\mathbf{S})) - F_{\mathcal{D}}(x^*).$$

Note that $\varepsilon_{\text{risk}}(\mathcal{A})$ is a random variable (due to randomness in the sample $\mathbf{S}$ and any possible internal randomness of the algorithm). Our guarantees on the excess population risk will be expressed in terms of upper bounds on this quantity that hold *with high probability* over the randomness of both $\mathbf{S}$ and the random coins of the algorithm.

**Empirical risk minimization (ERM)** is one of the most standard approaches to stochastic convex optimization. In the ERM problem, we are given a sample $\mathbf{S} = (\mathbf{z}_1, \ldots, \mathbf{z}_n)$, and the goal is to find

$$x^*(\mathbf{S}) \in \arg\min\left\{F_{\mathbf{S}}(x) := \frac{1}{n}\sum_{i=1}^{n} f(x, \mathbf{z}_i) : x \in \mathcal{X}\right\}.$$

One way to bound the excess population risk is to solve the ERM problem, and appeal to uniform convergence; however, uniform convergence rates in this case are dimension-dependent, $\Omega(\sqrt{d/n})$ [16].

**Risk decomposition:** Guaranteeing low excess population risk for a general algorithm is a nontrivial task. A common way to bound it is by decomposing it into *generalization*, *optimization* and *approximation error*:

$$\varepsilon_{\text{risk}}(\mathcal{A}) \leq \underbrace{F_{\mathcal{D}}(\mathcal{A}(\mathbf{S})) - F_{\mathbf{S}}(\mathcal{A}(\mathbf{S}))}_{\varepsilon_{\text{gen}}(\mathcal{A})} + \underbrace{F_{\mathbf{S}}(\mathcal{A}(\mathbf{S})) - F_{\mathbf{S}}(x^*(\mathbf{S}))}_{\varepsilon_{\text{opt}}(\mathcal{A})} + \underbrace{F_{\mathbf{S}}(x^*(\mathbf{S})) - F_{\mathcal{D}}(x^*)}_{\varepsilon_{\text{approx}}}. \qquad (3)$$

Here, the optimization error corresponds to the empirical optimization gap, which can be bounded by standard optimization convergence analysis. The expected value of the approximation error is at most zero. One can show, e.g., by Hoeffding's inequality, that the approximation error is bounded by $\tilde{O}(LR/\sqrt{n})$ with high probability (see Lemma 2.1 below.) Therefore, to establish bounds on the excess risk it suffices to upper bound the optimization and generalization errors.

**Lemma 2.1.** *For any $\theta \in (0,1)$, with probability at least $1 - \theta$, the approximation error is bounded as*

$$\varepsilon_{\text{approx}} \leq \frac{RL\sqrt{2\log(1/\theta)}}{\sqrt{n}}.$$

*Proof.* First, note that $F_{\mathcal{D}}(x^*) = \mathbb{E}_{\mathbf{S}}[F_S(x^*)] = \frac{1}{n}\sum_{i=1}^{n} f(x^*, \mathbf{z}_i)$. Hence, by independence and the fact that $f(x^*, \mathbf{z}_i) \in [-RL, RL]$ with probability 1 for all $i \in [n]$, the following follows from Hoeffding's inequality:

$$\mathbb{P}_{\mathbf{S}\sim\mathcal{D}^n}\left[F_{\mathbf{S}}(x^*) - F_{\mathcal{D}}(x^*) \geq \frac{RL\sqrt{2\log(1/\theta)}}{\sqrt{n}}\right] \leq \theta.$$

Finally, note that by definition of $x^*(\mathbf{S})$, we have $F_{\mathbf{S}}(x^*(\mathbf{S})) - F_{\mathbf{S}}(x^*) \leq 0$. Combining this with the above bound completes the proof. $\qquad \square$

We say that two datasets $S, S'$ are neighboring, denoted $S \simeq S'$, if they only differ on a single entry; i.e., there exists $i \in [n]$ s.t. for all $k \neq i$, $z_k = z'_k$.

**Uniform argument stability (UAS):** Given an algorithm $\mathcal{A}$ and datasets $S \simeq S'$, we define the *uniform argument stability* (UAS) random variable as

$$\delta_{\mathcal{A}}(S, S') := \|\mathcal{A}(S) - \mathcal{A}(S')\|.$$

The randomness here is due to any possible internal randomness of $\mathcal{A}$. For any $L$-Lipschitz function $f$, we have that $f(\mathcal{A}(S), z) - f(\mathcal{A}(S'), z) \leq L\,\delta_{\mathcal{A}}(S, S')$. Hence, upper bounds on UAS can be easily transformed into upper bounds on uniform stability.

In this work, we will consider two types of bounds on UAS.

## 2.1 High-probability guarantees on UAS

In Section 3, we give upper bounds on UAS for three variants of the (stochastic) gradient descent algorithm, namely, (i) full-batch gradient descent, (ii) sampling-with-replacement stochastic gradient descent, and (ii) fixed-permutation stochastic gradient descent. Variant (i) is deterministic (and hence UAS is a deterministic quantity). For variant (ii), for any pair of neighboring datasets $S, S'$, we give an upper bound on the UAS random variable that holds with high probability over the algorithm's internal randomness (the sampling

with replacement). For variant (iii), we give an upper bound on UAS that holds for an arbitrary choice of permutation; in particular, for any random permutation our upper bound on the UAS random variable that holds with probability 1.

High-probability upper bounds on UAS lead to high-probability upper bounds on generalization error $\varepsilon_{\text{gen}}$. We will use the following theorem, which follows in a straightforward fashion from [20, Theorem 1.1], to derive generalization-error guarantees for our results in Sections 5 and 6 based on our UAS upper bounds in Section 3.

**Theorem 2.2** (follows from Theorem 1.1 in [20]). *Let $\mathcal{A} : \mathcal{Z}^n \to \mathcal{X}$ be a randomized algorithm. For any pair of neighboring datasets $S, S'$, suppose that the UAS random variable of $\mathcal{A}$ satisfies:*

$$\mathbb{P}_{\mathcal{A}} \left[ \delta_{\mathcal{A}}(S, S') \geq \gamma \right] \leq \theta_0.$$

*Then there is a constant $c$ such that for any distribution $\mathcal{D}$ over $\mathcal{Z}$ and any $\theta \in (0, 1)$, we have*

$$\mathbb{P}_{\mathbf{S} \sim \mathcal{D}^n, \mathcal{A}} \left[ |\varepsilon_{\text{gen}}(\mathcal{A})| \geq c \left( L\gamma \log(n) \log(n/\theta) + LR\sqrt{\frac{\log(1/\theta)}{n}} \right) \right] \leq \theta + \theta_0,$$

*where $\varepsilon_{\text{gen}}(\mathcal{A}) = F_{\mathcal{D}}(\mathcal{A}(\mathbf{S})) - F_{\mathbf{S}}(\mathcal{A}(\mathbf{S}))$ as defined earlier.*

## 2.2 Expectation guarantees on UAS

Our results also include upper and lower bounds on $\sup_{S \simeq S'} \mathbb{E}_{\mathcal{A}} [\delta_{\mathcal{A}}(S, S')]$; that is the supremum of the expected value of the UAS random variable, where the supremum is taken over all pairs of neighboring datasets. In Section 3.3.1, we provide an upper bound on this quantity for the sampling-with-replacement stochastic gradient descent. The upper bounds on the other two variants of the gradient descent method hold in the strongest sense (they hold with probability 1). Moreover, in Appendix A, we give slightly tighter expectation guarantees on UAS for both sampling-with-replacement SGD and fixed-permutation SGD with a uniformly random permutation.

In Section 4, we give lower bounds on this quantity for the two variants of the stochastic subgradient method, together with a deterministic lower bound for the full-batch variant.

# 3 Upper Bounds on Uniform Argument Stability

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

---

## 3.2 Upper Bounds for the Full Batch GD

As a direct corollary of Lemma 3.1, we derive the following upper bound on UAS for the batch gradient descent algorithm.

**Theorem 3.2.** *Let $\mathcal{X} \subseteq \mathcal{B}(0, R)$ and $\mathcal{F} = \mathcal{F}_{\mathcal{X}}^0(L)$. The full-batch gradient descent (Algorithm 1) has uniform argument stability*

$$\sup_{S \simeq S'} \delta_{\mathcal{A}_{\mathsf{GD}}}(S, S') \leq \min \left\{ 2R, \ 4L \left( \frac{1}{n} \sum_{t=1}^{T-1} \eta_t + \sqrt{\sum_{t=1}^{T-1} \eta_t^2} \right) \right\}.$$

*Proof.* The bound of $2R$ is obtained directly from the diameter bound on $\mathcal{X}$. Therefore, we focus exclusively on the second term. Let $S \simeq S'$ be arbitrary neighboring datasets, $x^1 = y^1$, and consider the trajectories $(x^t)_t, (y^t)_t$ associated with the batch GD method on datasets $S$ and $S'$, respectively. We use Lemma 3.1 with $f_t = F_S$ and $f_t' = F_{S'}$, for all $t \in [T-1]$. Notice that

$$\sup_{x \in \mathcal{X}} \| \nabla F_S(x) - \nabla F_{S'}(x) \| \leq 2L/n,$$

since $S \simeq S'$; in particular, $\| \nabla f_t(x^t) - \nabla f_t'(x^t) \| \leq a_t$, with $a_t = 2L/n$. We conclude by Lemma 3.1 that for all $t \in [T]$

$$\| x^t - y^t \| \leq 2L \sqrt{\sum_{s=1}^{t-1} \eta_s^2} + \frac{4L}{n} \sum_{s=2}^{t-1} \eta_s.$$

Hence, the stability bound holds for all the iterates, and thus for $\overline{x}^T$ by the triangle inequality. $\qquad\square$

## 3.3 Upper Bounds for SGD

Next, we state and prove upper bounds on UAS for two variants of stochastic gradient descent: sampling-with-replacement SGD (Section 3.3.1) and fixed-permutation SGD (Section 3.3.2). Here, we give strong upper bounds that hold with high probability (for sampling-with-replacement SGD) and with probability 1 (for fixed-permutation SGD). In Appendix A, we derive tighter upper bounds for these two variants of SGD in the case where the number of iterations $T <$ the number of samples in the data set $n$; however, the bounds derived in this case hold only in expectation.

### 3.3.1 Sampling-with-replacement SGD

Next, we study the uniform argument stability of the sampling-with-replacement stochastic gradient descent (Algorithm 2). This algorithm has the benefit that each iteration is extremely cheap compared to Algorithm 1. Despite these savings, we will show that same bound on UAS holds with high probability.

**Algorithm 2** $\mathcal{A}_{\text{rSGD}}$: Sampling with replacement SGD

---
**Require:** Dataset: $S = (z_1, \ldots, z_n) \in \mathcal{Z}^n$, # iterations $T$, stepsizes $\{\eta_t : t \in [T]\}$
  1: Choose arbitrary initial point $x^1 \in \mathcal{X}$
  2: **for** $t = 1$ to $T - 1$ **do**
  3:     Sample $\mathbf{i}_t \sim \text{Unif}([n])$
  4:     $x^{t+1} := \text{Proj}_{\mathcal{X}} \left( x^t - \eta_t \cdot \nabla f(x^t, z_{\mathbf{i}_t}) \right)$
  5: **return** $\overline{x}^T = \frac{1}{\sum_{t \in [T]} \eta_t} \sum_{t \in [T]} \eta_t x^t$

---

We now state and prove our upper bound for sampling-with-replacement SGD.

**Theorem 3.3.** *Let $\mathcal{X} \subseteq \mathcal{B}(0, R)$ and $\mathcal{F} = \mathcal{F}^0_{\mathcal{X}}(L)$. The uniform argument stability of the sampling-with-replacement SGD (Algorithm 2) satisfies:*

$$\sup_{S \simeq S'} \mathop{\mathbb{E}}_{\mathcal{A}_{\text{rSGD}}} \left[ \delta_{\mathcal{A}_{\text{rSGD}}}(S, S') \right] \leq \min \left( 2R, \; 4L \left( \sqrt{\sum_{t=1}^{T-1} \eta_t^2} + \frac{1}{n} \sum_{t=1}^{T-1} \eta_t \right) \right).$$

*Moreover, if $\eta_t = \eta > 0 \; \forall t$ then, for any pair $(S, S')$ of neighboring datasets, with probability at least $1 - \exp(-n/2)$ (over the algorithm's internal randomness), the UAS random variable is bounded as*

$$\delta_{\mathcal{A}_{\text{rSGD}}}(S, S') \leq \min \left( 2R, \; 4L \left( \eta \sqrt{T - 1} + \eta \frac{T-1}{n} \right) \right).$$

*Proof.* The bound of $2R$ trivially follows from the diameter bound on $\mathcal{X}$. We thus focus on the second term of the bound. Let $S \simeq S'$ be arbitrary neighboring datasets, $x^0 = y^0$, and consider the trajectories $(x^t)_{t \in [T]}, (y^t)_{t \in [T]}$ associated with the sampled-with-replacement stochastic subgradient method on datasets $S$ and $S'$, respectively. We use Lemma 3.1 with $f_t(\cdot) = f(\cdot, \mathbf{z}_{\mathbf{i}_t})$ and $f'_t(\cdot) = f(\cdot, \mathbf{z}_{\mathbf{i}_{t'}})$. Let us define $\mathbf{r}_t \triangleq \mathbf{1}_{\{\mathbf{z}_{\mathbf{i}_t} \neq \mathbf{z}'_{\mathbf{i}_t}\}}$. Note that at every step $t$, $\mathbf{r}_t = 1$ with probability $1 - 1/n$, and $\mathbf{r}_t = 0$ otherwise. Moreover, note that $\{\mathbf{r}_t : t \in [T]\}$ is an independent sequence of Bernoulli random variables. Finally, note that $\|\nabla f_t(x^t) - \nabla f'_t(x^t)\| \leq 2L\mathbf{r}_t$.

Hence, by Lemma 3.1, for any realization of the trajectories of the SGD method, we have

$$\forall t \in [T]: \quad \|x^t - y^t\| \leq 2L \sqrt{\sum_{s=1}^{t-1} \eta_s^2} + 4L \sum_{s=1}^{t-1} \mathbf{r}_s \eta_s \leq \Delta_T, \tag{5}$$

where $\Delta_T \triangleq 2L \sqrt{\sum_{s=1}^{T-1} \eta_s^2} + 4L \sum_{s=1}^{T-1} \mathbf{r}_s \eta_s$. Taking expectation of (5), we have

$$\forall t \in [T]: \quad \mathbb{E}\left[\|x^t - y^t\|\right] \leq \mathbb{E}[\Delta_T] = 2L \sqrt{\sum_{s=1}^{T-1} \eta_s^2} + \frac{4L}{n} \sum_{s=1}^{T-1} \eta_s.$$

This establishes the upper bound on UAS but only in expectation. Now, we proceed to prove the high-probability bound. Here, we assume that the step size is fixed; that is, $\eta_t = \eta > 0$ for all $t \in [T-1]$. Note that each $\mathbf{r}_s, s \in [T]$, has variance $\frac{1}{n}\left(1 - \frac{1}{n}\right) < \frac{1}{n}$. Hence, by Chernoff's bound[3], we have

$$\mathbb{P}\left[ \eta \sum_{s=1}^{T-1} \mathbf{r}_s \geq \eta \frac{T-1}{n} + \eta \sqrt{T-1} \right] \leq \exp\left( -\frac{\eta^2 (T-1)}{2\eta^2 \frac{T-1}{n}} \right) = \exp\left( -\frac{n}{2} \right).$$

Therefore, with probability at least $1 - \exp(-n/2)$, we have

$$\Delta_T \leq 3L\eta\sqrt{T-1} + \frac{4L}{n}\eta(T-1).$$

Putting this together with (5), with probability at least $1 - \exp(-n/2)$, we have

$$\forall t \in [T]: \quad \|x^t - y^t\| \leq 3L\eta\sqrt{T-1} + \frac{4L}{n}\eta(T-1).$$

Finally, by the triangle inequality, we get that with probability at least $1 - \exp(-n/2)$, the same stability bound holds for the average of the iterates $\overline{x}^T, \overline{y}^T$. □

### 3.3.2 Upper Bounds for the Fixed Permutation SGD

In Algorithm 3, we describe the fixed-permutation stochastic gradient descent. This algorithm works in epochs, where each epoch is a single pass on the data. The order in which data is used is the same across epochs, and is given by a permutation $\pi$. The algorithm can be alternatively described without the epoch loop simply by

$$x^{t+1} = \mathsf{Proj}_{\mathcal{X}}\left(x^t - \eta_t \cdot \nabla f(x^t, z_{\boldsymbol{\pi}(t \bmod n)})\right) \qquad (\forall t \in [nK]). \tag{6}$$

We will use this description for stability analysis, since it is more convenient.

---

**Algorithm 3** $\mathcal{A}_{\mathsf{PerSGD}}$: Fixed Permutation SGD
---

**Require:** Dataset $S = (z_1, \ldots, z_n) \in \mathcal{Z}^n$, # rounds $K$, total # steps $T \triangleq nK$, step sizes, $\{\eta_t\}_{t \in [nK]}$
    $\pi : [n] \to [n]$ permutation over $[n]$
1:   Choose arbitrary initial point $x_{n+1}^0 \in \mathcal{X}$
2:   **for** $k = 1, \ldots, K$ **do**
3:     $x_1^k = x_{n+1}^{k-1}$
4:     **for** $t = 1$ to $n$ **do**
5:       $x_{t+1}^k := \mathsf{Proj}_{\mathcal{X}}\left(x_t^k - \eta_{(k-1)n+t} \cdot \nabla f(x_t^k, z_{\pi(t)})\right)$
6:     $\overline{\eta}_k = \sum_{t=1}^n \eta_{(k-1)n+t}$
7:   **return** $\overline{x}^K = \frac{1}{\sum_{k \in [K]} \overline{\eta}_k} \sum_{k \in [K]} \overline{\eta}_k \cdot x_1^k$

---

We show that the same UAS bound of batch gradient descent and sampling-with-replacement SGD holds for the fixed-permutation SGD. We also observe that a slightly tighter bound can be achieved if we consider *the expectation guarantee* on UAS when $\boldsymbol{\pi}$ is chosen uniformly at random. We leave these details to Theorem A.2 in the Appendix.

In the next result, we assume that the sequence of step sizes $(\eta_t)_{t \in [T]}$ is non-increasing, which is indeed the case for almost all known variants of SGD.

**Theorem 3.4.** *Let $\mathcal{X} \subseteq \mathcal{B}(0, R)$, $\mathcal{F} = \mathcal{F}_{\mathcal{X}}^0(L)$, and $\pi$ be any permutation over $[n]$. Suppose the step sizes $(\eta_t)_{t \in [T]}$ form a non-increasing sequence. Then the uniform argument stability of the fixed-permutation SGD (Algorithm 3) is bounded as*

$$\sup_{S \simeq S'} \delta_{\mathcal{A}_{\mathsf{PerSGD}}}(S, S') \leq \min\left\{2R, \ 2L\left(\sqrt{\sum_{t=1}^{T-1} \eta_t^2} + \frac{2}{n}\sum_{t=1}^{T-1} \eta_t\right)\right\}.$$

*Proof.* Again, the bound of $2R$ is trivial. Now, we show the second term of the bound. Let $S \simeq S'$ be arbitrary neighboring datasets, $x^1 = y^1$, and consider the trajectories $(x^t)_{t \in [T]}, (y^t)_{t \in [T]}$ associated with the fixed permutation stochastic subgradient method on datasets $S$ and $S'$, respectively. Since the datasets $S \simeq S'$ are arbitrary, we may assume without loss of generality that $\pi$ is the identity, whereas the perturbed coordinate $\mathbf{i} = i$ is arbitrary. We use Lemma 3.1 with $f_t(\cdot) = f\left(\cdot, \mathbf{z}_{(t \bmod n)}\right)$ and $f'_t(\cdot) = f\left(\cdot, \mathbf{z}'_{(t \bmod n)}\right)$. It is easy to see then that $\|\nabla f_t(x^t) - \nabla f'_t(x^t)\| \le a_t$, with $a_t = 2L \cdot \mathbf{1}_{\{(t \bmod n)=i\}}$, where $\mathbf{1}_{\{\text{condition}\}}$ is the indicator of condition. Hence, by Lemma 3.1, we have

$$
\begin{aligned}
\|x^t - y^t\| &\le 2L\sqrt{\sum_{s=1}^{t-1} \eta_s^2} + 4L \sum_{r=1}^{\lfloor (t-1)/n \rfloor} \eta_{rn+i} \\
&\le 2L\sqrt{\sum_{s=1}^{t-1} \eta_s^2} + \frac{4L}{n} \sum_{r=1}^{t-1} \eta_s,
\end{aligned}
$$

where at the last step we used the fact that $(\eta_t)_{t \in [T]}$ is non-increasing; namely, for any $r \ge 1$

$$
\eta_{rn+i} \le \frac{1}{n} \sum_{s=(r-1)n+i+1}^{rn+i} \eta_s.
$$

Since the bound holds for all the iterates, using triangle inequality, it holds for the output $\overline{x}^K$ averaged over the iterates from the $T/n$ epochs. $\qquad\square$

## 3.4 Discussion of the upper bounds: examples of specific instantiations

The upper bounds on stability from this section all behave very similarly. Let us explore the consequences of the obtained rates in terms of generalization bounds for different choices of the step size sequence. As a case study, we will consider excess risk bounds for the full-batch subgradient method (Algorithm 1), but similar conclusions hold for all the variants that we studied. We emphasize that prior to this work, no dimension-independent bounds on the excess risk were known of this method (specifically, for nonsmooth losses and without explicit regularization).

To bound the excess risk, we will use the risk decomposition, eqn. (3). For simplicity, we will only be studying excess risk bounds in expectation (in Section 6 we consider stronger, high probability, bounds). In this case, the stability implies generalization result (Theorem 2.2) simplifies to [5, 21]

$$
\mathbb{E}_{\mathbf{S}}[F_{\mathbf{S}}(\mathcal{A}(\mathbf{S})) - F_{\mathcal{D}}(\mathcal{A}(\mathbf{S}))] \le \sup_{S \simeq S'} \delta_{\mathcal{A}}(S, S').
$$

Finally, the approximation error (Lemma 2.1) simplifies as well: it is upper bounded by 0 in expectation.

- *Fixed stepsize:* Let $\eta_t \equiv \eta > 0$. By Thm. 3.2, UAS is bounded by $4L\sqrt{T}\eta + \frac{4LT\eta}{n}$. On the other hand, the standard analysis of subgradient descent guarantees that $\varepsilon_{\text{opt}}(\mathcal{A}_{\text{GD}}) \le \frac{R^2}{2\eta T} + \frac{\eta L^2}{2}$. Therefore, by the expected risk decomposition (3)

$$
\mathbb{E}_{\mathbf{S}}[\varepsilon_{\text{risk}}(\mathcal{A}_{\text{GD}})] \le \mathbb{E}_{\mathbf{S}}[\varepsilon_{\text{gen}}(\mathcal{A}_{\text{GD}})] + \mathbb{E}_{\mathbf{S}}[\varepsilon_{\text{opt}}(\mathcal{A}_{\text{GD}})] \le 4L^2\sqrt{T}\eta + \frac{4L^2 T\eta}{n} + \frac{R^2}{2\eta T} + \frac{\eta L^2}{2}.
$$

If we consider the standard method choice, $\eta = R/[L\sqrt{n}]$ and $T = n$, the bound above is at least $4LR$ (due to the first term). Consequently, upper bounds obtained from this approach are vacuous.

In order to deal with the $L\sqrt{T}\eta$ term, we need to substantially moderate our stepsize, together with running the algorithm for longer. For example, $\eta = \frac{R}{4L\sqrt{Tn}}$ gives $\mathbb{E}_{\mathbf{S}}[\varepsilon_{\text{risk}}(\mathcal{A}_{\text{GD}})] \leq \frac{2LR}{\sqrt{n}} + \frac{2LR\sqrt{n}}{\sqrt{T}} + \frac{R\sqrt{T}}{n^{3/2}}$, so by choosing $T = n^2$ we obtain an expected excess risk bound of $O(LR/\sqrt{n})$, which is optimal. We will see next that it is not possible to obtain the same rates from this bound if $T = o(n^2)$, for any choice of $\eta > 0$. It is also an easy observation that, at least for constant stepsize, it is not possible to recover the optimal excess risk if $T = \omega(n^2)$.

- *Varying stepsize:* For a general sequence of stepsizes the optimization guarantees of Algorithm 1 are the following

$$\mathbb{E}_{\mathbf{S}}[\varepsilon_{\text{opt}}(\mathcal{A}_{\text{GD}})] \leq \frac{R^2}{2\sum_{t=1}^{T-1}\eta_t} + \frac{L^2\sum_{t=1}^{T-1}\eta_t^2}{2}.$$

From the risk decomposition, we have

$$\begin{aligned}
\mathbb{E}_{\mathbf{S}}[\varepsilon_{\text{risk}}(\mathcal{A}_{\text{GD}})] &\leq \mathbb{E}_{\mathbf{S}}[\varepsilon_{\text{gen}}(\mathcal{A}_{\text{GD}})] + \mathbb{E}_{\mathbf{S}}[\varepsilon_{\text{opt}}(\mathcal{A}_{\text{GD}})] \\
&\leq 4L^2\sqrt{\sum_{t=1}^{T-1}\eta_t^2} + \frac{4L^2}{n}\sum_{t=1}^{T-1}\eta_t + \frac{R^2}{2\sum_{t=1}^{T-1}\eta_t} + \frac{L^2\sum_{t=1}^{T-1}\eta_t^2}{2}.
\end{aligned}$$

In fact, we can show that any choice of step sizes that makes the quantity above $O(LR/\sqrt{n})$ must necessarily have $T = \Omega(n^2)$. Indeed, notice that in such case

$$\frac{R^2}{2\sum_{t=1}^{T-1}\eta_t} = O\Big(\frac{LR}{\sqrt{n}}\Big); \quad 4L^2\sqrt{\sum_{t=1}^{T-1}\eta_t^2} = O\Big(\frac{LR}{\sqrt{n}}\Big)$$

$$\iff \quad \sum_{t=1}^{T-1}\eta_t = \Omega\Big(\frac{R\sqrt{n}}{L}\Big); \quad \sqrt{\sum_{t=1}^{T-1}\eta_t^2} = O\Big(\frac{R}{L\sqrt{n}}\Big).$$

Therefore, by Cauchy-Schwarz inequality,

$$\Omega\Big(\frac{R\sqrt{n}}{L}\Big) = \sum_t \eta_t \leq \sqrt{T}\sqrt{\sum_t \eta_t^2} = O\Big(\frac{R\sqrt{T}}{L\sqrt{n}}\Big) \quad \implies \quad T = \Omega(n^2).$$

The high iteration complexity required to obtain optimal bounds motivates studying whether it is possible to improve our uniform argument stability bounds. We will show that, unfortunately, they are sharp up to absolute constant factors.

# 4 Lower Bounds on Uniform Argument Stability

In this section we provide matching lower bounds for the previously studied first-order methods. These lower bounds show that our analyses are tight, up to absolute constant factors.

We note that it is possible to prove a general purpose lower bound on stability by appealing to sample complexity lower bounds for stochastic convex optimization [34]. This approach in the smooth convex case was first studied in [10]; there, these lower bounds are sharp. However, in the nonsmooth case they are very far from bounds in the previous section. The idea is that for sufficiently small step size, a first-order method must incur $\Omega(LT\eta/n)$ uniform stability.

**Observation 4.1.** *Let $\mathcal{A}$ be a $\gamma$-uniformly stable stochastic convex optimization algorithm with $\gamma = s(T)/n$, where $s(T)$ is increasing and $\lim_{T \to +\infty} s(T) = +\infty$. By the lower bound on the optimal risk of nonsmooth convex optimization, $\varepsilon_{risk} \geq \frac{LR}{C_1 \sqrt{n}}$, where $C_1 > 0$ is a universal constant [34]. This, combined with the risk decomposition* (3), *implies that*

$$\varepsilon_{opt} \geq \frac{LR}{C_1 \sqrt{n}} - \frac{s(T)}{n} = -s(T)\Big(\frac{1}{\sqrt{n}} - \frac{LR}{2C_1 s(T)}\Big)^2 + \frac{(LR)^2}{4C_1^2 s(T)}.$$

*By our assumption on $s(T)$, for $T$ sufficiently large, there always exists $n$ such that*

$$\frac{4C_1 s(T)}{3LR} \leq \sqrt{n} \leq \frac{4C_1 s(T)}{LR}$$

*which leads to $\varepsilon_{opt} \geq \frac{(LR)^2}{C_2 s(T)}$, where $C_2 > 0$ is a universal constant.*

*If algorithm $\mathcal{A}$ is based on $T$ subgradient iterations with constant step size $\eta > 0$ (these could be either stochastic, batch or minibatch), by standard analysis, the optimization guarantee of such algorithm is $\varepsilon_{opt} \leq \frac{1}{2}(\frac{R^2}{\eta T} + \eta L^2)$. Both bounds in combination give*

$$s(T) \geq \frac{2(LR)^2}{C_2(\eta L^2 + R^2/[\eta T])} = \frac{2(LR)^2 \eta T}{C_2(\eta^2 T L^2 + R^2)}.$$

*If we further assume that $\eta \leq (R/L)/\sqrt{T}$ (notice $\eta = (R/L)/\sqrt{T}$ minimizes the optimization error), then $s(T) \geq L^2 \eta T/C_2$. We also emphasize all the choices of step size that we will make to control generalization error will lie in this range.*

This reasoning leads to an $\Omega(LT\eta/n)$ lower bound on uniform argument stability, that can be added to any other lower bound we can prove on specific algorithms that enjoy rates as of gradient descent.

Next we will prove finer lower bounds on the UAS of specific algorithms. For this, note that the objective functions we use are polyhedral, thus the subdifferential is a polytope at any point. Since the algorithm should work for any oracle, we will let the subgradients provided to be extreme points, $\nabla f(x, z) \in \text{ext}(\partial f(x, z))$. Moreover, we can make adversarial choices of the chosen subgradient.

## 4.1 Lower Bounds for Full Batch GD

**Theorem 4.2.** *Let $\mathcal{X} = \mathcal{B}(0, 1)$, $\mathcal{F} = \mathcal{F}_{\mathcal{X}}^0(1)$ and $d \geq \min\{T, 1/\eta^2\}$. For the full-batch gradient descent (Alg. 1) with constant step size $\eta > 0$, there exist $S \simeq S'$ such that the UAS is lower bounded as $\delta_{\mathcal{A}_{\text{GD}}(S,S')} = \Omega(\min\{1, \eta\sqrt{T} + \eta T/n\})$.*

*Proof.* Let $D \triangleq \min\{T, 1/\eta^2\} \leq d$, and $\nu, K > 0$. We consider $\mathcal{Z} = \{0, 1\}$, and the objective function

$$f(x, z) = \begin{cases} \max\{0, x_1 - \nu, \ldots, x_D - \nu\} & \text{if } z = 0 \\ \langle r, x \rangle / K & \text{if } z = 1, \end{cases}$$

where $r = (-1, \ldots, -1, 0, \ldots, 0)$ (i.e., supported on the first $D$ coordinates). Notice that for normalization purposes, we need $K \geq \sqrt{D}$; furthermore, we will choose $K$ sufficiently large such that $T\sqrt{D}/[nK] = o(1)$. Consider the data sets $S \simeq S'$, leading to the empirical objectives:

$$F_S(x) = \frac{1}{nK}\langle r, x \rangle + \frac{n-1}{n} \max\{0, x_1 - \nu, \ldots, x_D - \nu\} \text{ and } F_{S'}(x) = \max\{0, x_1 - \nu, \ldots, x_D - \nu\}.$$

Let $(x^t)_{t\in[T]}$ and $(y^t)_{t\in[T]}$ be the trajectories of the algorithm over datasets $S$ and $S'$, respectively, initialized from $x^1 = y^1 = 0$. Clearly, $y^t = 0$ for all $t$. Now $x^2 = -\frac{\eta}{nK} r$; choosing $\nu < \eta/(nK)$, we have $\nabla f(x^2, z) = -\frac{\eta}{nK} r + \frac{n-1}{n} e_1$, and hence $x^3 = -\frac{2\eta}{nK} r - \eta \frac{n-1}{n} e_1$. Sequentially, the method will perform cumulative subgradient steps on $e_2, e_3 \ldots, e_D$. In particular, for any $t \in [D+1]$, we have $x^{t+1} = -t\frac{\eta}{nK} r - \eta \frac{n-1}{n} \sum_{s=1}^{t-1} e_s$.

By orthogonality of the subgradients and given our choice of $K$, we conclude that

$$
\begin{aligned}
\|x^{D+2} - y^{D+2}\| &= \|x^{D+2}\| \geq \frac{\eta}{2} \Big\| \sum_{t=1}^{D} e_t \Big\| - \eta \frac{D\sqrt{D}}{nK} \\
&\geq \frac{\eta}{2}\sqrt{D} - \eta \cdot o(1) = \Omega(\eta\sqrt{D}) \\
&= \Omega(\min\{1, \eta\sqrt{T}\}),
\end{aligned}
$$

and further subgradient steps $t = D+1, \ldots, T$ are only given by the linear term, $r/[nK]$, which are negligible perturbations.

We finish by arguing that averaging does not help. First, in the case $D = T$:

$$
\begin{aligned}
\delta(\mathcal{A}_{\mathsf{GD}}) &= \|\bar{x}^T\| \geq \frac{\eta}{2} \Big\| \frac{1}{T} \sum_{t=1}^{T} \sum_{s=1}^{t-2} e_s \Big\| - o(1) = \frac{\eta}{2} \Big\| \sum_{s=1}^{T} \frac{T-s-2}{T} e_s \Big\| - o(1) \\
&\geq \frac{\eta}{4} \Big\| \sum_{s \leq T/2-2} e_s \Big\| - o(1) = \Omega(\eta\sqrt{T}).
\end{aligned}
$$

And second, in the case $D = 1/\eta^2$:

$$
\begin{aligned}
\delta(\mathcal{A}_{\mathsf{GD}}) &= \|\bar{x}^T\| \geq \frac{\eta}{2T} \Big\| \sum_{t=1}^{D+2} \sum_{s=1}^{t-2} e_s + \sum_{t=D+3}^{T} \sum_{s=1}^{D} e_s \Big\| - o(1) \\
&= \frac{\eta}{2T} \Big\| \sum_{s=1}^{D} (D-s-1)e_s + \sum_{s=1}^{D} (T-D+2)e_s \Big\| - o(1) \\
&= \frac{\eta}{2} \Big\| \sum_{s=1}^{D-1} \frac{T-s+1}{T} e_s \Big\| - o(1) \geq \frac{\eta}{4} \Big\| \sum_{s=1}^{D/2} e_s \Big\| - o(1) = \Omega(\sqrt{D}\eta) = \Omega(1).
\end{aligned}
$$

Finally, the additional term $\Omega(\eta T/n)$ in the lower bound is obtained by Observation 4.1. $\qquad\square$

## 4.2 Lower Bounds for SGD Sampled with Replacement

We use a similar construction as from the previous result to prove a sharp lower bound on the uniform argument stability for stochastic gradient descent where the sampling is with replacement.

**Theorem 4.3.** *Let $\mathcal{X} = \mathcal{B}(0,1)$, $\mathcal{F} = \mathcal{F}_{\mathcal{X}}^0(1)$, and $d \geq \min\{T, 1/\eta^2\}$. For the sampled with replacement stochastic gradient descent (Algorithm 2) with constant step size $\eta > 0$, there exist $S \simeq S'$ such that the uniform argument stability satisfies $\mathbb{E}[\delta_{\mathcal{A}_{\mathsf{rSGD}}}(S, S')] = \Omega\Big( \min\big\{1, \frac{T}{n}\big\} \eta\sqrt{T} + \frac{\eta T}{n} \Big)$.*

*Proof.* Let $D \triangleq \min\{T, 1/\eta^2\} \leq d$, and $\nu > 0$, $K \geq \sqrt{D}$. Consider $\mathcal{Z} = \{0, 1\}$ and define

$$
f(x, z) = \begin{cases} \max\{0, x_1 - \nu, \ldots, x_D - \nu\} & \text{if } z = 0 \\ \langle r, x \rangle / K & \text{if } z = 1, \end{cases}
$$

where $r = (-1, \ldots, -1, 0, \ldots, 0)$ (i.e., supported on the first $D$ coordinates). Let the random sequence of indices used by the algorithm: $(\mathbf{i}_t)_{t \geq 0} \overset{i.i.d.}{\sim} \mathrm{Unif}([n])$. Let $S = (1, 0, \ldots, 0)$ and $S' = (0, 0, \ldots, 0)$ be neighboring datasets, and denote by $(x^t)_t$ and $(y^t)_t$ the respective stochastic gradient descent trajectories on $S$ and $S'$, initialized at $x^1 = y^1 = 0$. It is easy to see that under $S'$, we have $y^t = 0$ for all $t \in [T]$. Now, suppose that $\nu < \eta/K$. Then, we only have $x^t = 0$ for all $t \leq \tau$, where $\tau := \inf\{t \geq 1 : \mathbf{i}_t = 1\}$. After time $\tau$, $x^{\tau+1} = -\eta r/K$, and consequently $x^{\tau+1+j} = -\frac{\eta \mathbf{k}(\tau+j)}{K} r - \eta \sum_{s=1}^{j - \mathbf{k}(\tau+j)+1} e_s$, for all $j \in [D + \mathbf{k}(\tau+j) - 1]$, where $\mathbf{k}(t) \triangleq |\{s \in [t] : \mathbf{i}_s = 1\}|$. Note that conditioned on any fixed value for $\tau$, $\mathbf{k}(\tau + j) \leq j + 1$.

Let $\delta_T = \|x^T - y^T\| = \|x^T\|$. Hence, we have $\delta_T \geq \eta \| \sum_{s=1}^{T-\tau-\mathbf{k}(T-1)} e_s \| - \eta \mathbf{k}(T-1)\sqrt{D}/K \geq \eta\sqrt{T - \mathbf{k}(T-1) - \tau} - \eta T \sqrt{D}/K$. Let $\mathbf{k} = \mathbf{k}(T-1)$ from now on. Note that conditioned on any value for $\tau$, $\mathbf{k} - 1$ is a binomial random variable taking values in $\{0, \ldots, T-1-\tau\}$. Hence, conditioned on $\tau = t$, by the binomial tail, we always have $\mathbb{P}[\mathbf{k} > T/2 \mid \tau = t] \leq \exp(-T/4)$ for all $t \in [T]$ (in particular, this conditional probability is zero when $t \geq T/2$). Also, note that the same upper bound is valid without conditioning on $\tau$. Hence, by the law of total expectation, we have

$$\mathbb{E}[\delta_T] \;\;=\;\; \mathbb{E}[\delta_T | \, \mathbf{k} \leq T/2] \cdot \mathbb{P}[\mathbf{k} \leq T/2] + \mathbb{E}[\delta_T | \, \mathbf{k} > T/2] \cdot \mathbb{P}[\mathbf{k} > T/2] \geq c\,\mathbb{E}[\delta_T | \, \mathbf{k} \leq T/2]$$

where $c = (1 - \exp(-T/4)) = \Omega(1)$. Hence,

$$
\begin{aligned}
\mathbb{E}[\delta_T] \;\;\geq\;\; & c \sum_{t=1}^{T/2} \mathbb{E}[\delta_T | \tau = t, \, \mathbf{k} \leq T/2]\, \mathbb{P}[\tau = t | \, \mathbf{k} \leq T/2] \\
\geq\;\; & c^2 \sum_{t=1}^{T/2} \mathbb{E}[\delta_T | \tau = t, \, \mathbf{k} \leq T/2]\, \mathbb{P}[\tau = t] \\
\geq\;\; & c^2 \frac{\eta}{n} \sum_{t=1}^{T/2} \sqrt{T - T/2 - t}\,\Big(1 - \frac{1}{n}\Big)^{t-1} - c^2 \eta \sqrt{D} T/K.
\end{aligned}
$$

We choose $K$ sufficiently large such that $\eta\sqrt{D}T/K = o(\eta \min\{T^{3/2}/n, \sqrt{T}\})$. If $T \leq n$ then

$$\mathbb{E}[\delta_T] \geq c^2 \frac{\eta}{n} \sum_{t=1}^{T/2} \sqrt{t}\,\Big(1 - \frac{1}{n}\Big)^{n-2} - c^2 \frac{\eta\sqrt{D}T}{K} \geq c^2 \frac{\eta e^{-1}}{n} \sum_{t=1}^{T/2} \sqrt{t} - o\Big(\frac{\eta T^{3/2}}{n}\Big) = \Omega\Big(\frac{\eta T^{3/2}}{n}\Big)$$

and if $T > n$ then

$$\mathbb{E}[\delta_T] \geq c^2 \frac{\eta}{n} \sum_{t=1}^{n/4} \sqrt{T/2 - n/4}\, e^{-1} - o(\eta\sqrt{T}) = \Omega(\eta\sqrt{T})$$

This gives a lower bound on $\mathbb{E}[\delta_T]$. Proving that $\overline{x}^T$ satisfies the same lower bound is analogous to the proof in Theorem 4.2. Finally, $\Omega(\eta T/n)$ can be added to the lower bound by Observation 4.1. □

## 4.3 Lower Bounds for the Fixed Permutation Stochastic Gradient Descent

Finally, we study fixed permutation SGD.

**Theorem 4.4.** *Let $\mathcal{X} = \mathcal{B}(0, 1)$, $\mathcal{F} = \mathcal{F}_{\mathcal{X}}^0(1)$ and $d \geq \min\{T, 1/\eta^2\}$. For the fixed permutation stochastic gradient descent (Algorithm 3) with constant step size $\eta > 0$, there exist $S \simeq S'$ such that the uniform argument stability is lower bounded by $\mathbb{E}[\delta_{\mathcal{A}_{\mathsf{PerSGD}}}(S, S')] = \Omega\Big( \min\big\{1, \frac{T}{n}\big\} \eta\sqrt{T} + \frac{\eta T}{n}\Big)$.*

*Proof.* We consider the same function class of Thm. 4.2, and neighbor datasets $S' = (0, 0, \ldots, 0)$, $S = (1, 0, \ldots, 0)$. We will assume in what follows that $D = \min\{T, 1/\eta^2\}$, $K$ is sufficiently large and $\nu < \eta\|r\|/K$. Let $(x^t)_{t \in [T]}$ and $(y^t)_{t \in [T]}$ be the trajectories of Algorithm 3 over datasets $S, S'$ respectively, both initialized at $x^1 = y^1 = 0$. Let now $\tau = \boldsymbol{\pi}^{-1}(1) \sim \mathrm{Unif}[n]$. Arguing as in Thm. 4.2, we have that $y^t = 0$ for all $t$, whereas

$$
x^{t+1} = \begin{cases} 0 & t < \tau \\ -\frac{\eta(1+\lfloor t/n \rfloor)r}{K} - \eta \sum_{s=1}^{t-\tau-(1+\lfloor t/n \rfloor)} e_s & \tau \le t \le \tau + D. \end{cases}
$$

Later iterations will satisfy $\|x^t\| = 1 - o(1)$ if $D = 1/\eta^2$ (and otherwise the algorithm stops earlier). Therefore, for all $t \in [T]$,

$$
\begin{aligned}
\mathbb{E}_{\boldsymbol{\pi}}[\|x^t - y^t\|] &= \sum_{s=1}^{n} \mathbb{E}[\|x^t - y^t\| \mid \tau = s]\mathbb{P}[\tau = s] \\
&\ge \frac{\eta}{2n} \sum_{s=1}^{\min\{t,n\}} \sqrt{t-s} - \eta \cdot o(1) \\
&= \begin{cases} \Omega(\frac{\eta t^{3/2}}{n}) & \text{if } t \le n \\ \Omega(\eta\sqrt{t}) & \text{if } t > n. \end{cases}
\end{aligned}
$$

Notice that we used above that $K$ is such that $T\sqrt{D}/nK = o(1)$. Analogously as in Thm. 4.2, we can obtain the same conclusion for $\bar{x}^T$. The lower bound of $\eta T/n$ can be added by Observation 4.1, so the result follows. $\qquad\square$

# 5  Generalization Guarantees for Multi-pass SGD

One important implication of our stability bounds is that they provide non-trivial generalization error guarantees for multi-pass SGD on nonsmooth losses. Multi-pass SGD is one of the most extensively used settings of SGD in practice, where SGD is run for $K$ passes (epochs) over the dataset (namely, the number of iterations $T = Kn$). To the best of our knowledge, aside from the dimension-dependent bounds based on uniform convergence [39], no generalization error guarantees are known for the multi-pass setting on general nonsmooth convex losses. Given our uniform stability upper bounds, we can prove the following generalization error guarantees for the multi-pass setting of sampling-with-replacement SGD. Analogous results can be obtained for fixed-permutation SGD .

**Theorem 5.1.** *Running Algorithm 2 for $K$ passes (i.e., for $T = Kn$ iterations) with constant stepsize $\eta_t = \eta > 0$ yields the following generalization error guarantees:*

$$
|\mathbb{E}_{\mathcal{A}_{\mathsf{rSGD}}}[\varepsilon_{\mathrm{gen}}(\mathcal{A}_{\mathsf{rSGD}})]| \le 4L^2\eta\left(\sqrt{Kn} + K\right),
$$

*and there exists $c > 0$, such that for any $0 < \theta < 1$, with probability $\ge 1 - \theta - \exp(-n/2)$,*

$$
|\varepsilon_{\mathrm{gen}}(\mathcal{A}_{\mathsf{rSGD}})| \le c\left(L^2\eta\left(\sqrt{Kn} + K\right)\log(n)\log(n/\theta) + LR\sqrt{\frac{\log(1/\theta)}{n}}\right).
$$

*Proof.* First, by the expectation guarantee on UAS given in Theorem 3.3 together with the fact that the losses are $L$-Lipschitz, it follows that Algorithm 2 (when run for $K$ passes with constant stepsize $\eta$) is $\gamma$-uniformly stable, where $\gamma = 4L^2\left(\eta\sqrt{Kn} + \eta K\right)$. Then, by [21, Thm. 2.2], we have

$$\left|\mathbb{E}_{\mathcal{A}_{\mathsf{rSGD}}}[\varepsilon_{\mathrm{gen}}(\mathcal{A}_{\mathsf{rSGD}})]\right| \leq \gamma.$$

For the high-probability bound, we combine the high-probability guarantee on UAS given in Theorem 3.3 with Theorem 2.2 to get the claimed bound. □

These bounds on generalization error can be used to obtain excess risk bounds using the standard risk decomposition (see (3)). In practical scenarios where one can certify small optimization error for multi-pass SGD, Thm. 5.1 can be used to readily estimate the excess risk. In Section 6.2 we provide worst-case analysis showing that multi-pass SGD is guaranteed to attain the optimal excess risk of $\approx LR/\sqrt{n}$ within $n$ passes (with appropriately chosen constant stepsize).

# 6 Implications of Our Stability Bounds

## 6.1 Differentially Private Nonsmooth Stochastic Convex Optimization

Now we show an application of our stability upper bound to *differentially private* stochastic convex optimization (DP-SCO). Here, the input sample to the stochastic convex optimization algorithm is a sensitive and private data set, thus the algorithm is required to satisfy the notion of $(\alpha, \beta)$-differential privacy. A randomized algorithm $\mathcal{A}$ is $(\alpha, \beta)$-differentially private if, for any pair of datasets $S \simeq S'$, and for all events $\mathcal{O}$ in the output range of $\mathcal{A}$, we have

$$\mathbb{P}\left[\mathcal{A}(S) \in \mathcal{O}\right] \leq e^{\alpha} \cdot \mathbb{P}\left[\mathcal{A}(S') \in \mathcal{O}\right] + \beta,$$

where the probability is taken over the random coins of $\mathcal{A}$ [15, 14]. For meaningful privacy guarantees, the typical settings of the privacy parameters are $\alpha < 1$ and $\beta \ll 1/n$.

Using our UAS upper bounds, we show that a simple variant of noisy SGD [3], that requires only $n^2$ gradient computations, yields the optimal excess population risk for DP-SCO. In terms of running time, this is a small improvement over the algorithm of [17] for the nonsmooth case, which requires $O(n^2\sqrt{\log 1/\beta})$ gradient computations. More importantly, our algorithm is substantially simpler. For comparison, the algorithm in [17] is based on a multi-phase SGD, where in each phase a separate regularized ERM problem is solved. To ensure privacy, the output of each phase is perturbed with an appropriately chosen amount of noise before being used as the initial point for the next phase.

The description of the algorithm is given in Algorithm 4.

**Theorem 6.1** (Privacy guarantee of $\mathcal{A}_{\mathsf{NSGD}}$). *Algorithm 4 is $(\alpha, \beta)$-differentially private.*

The proof of the theorem follows the same lines of [3, Theorem 2.1], but we replace their privacy analysis of the Gaussian mechanism with the tighter Moments Accountant method of [1]. analysis of [1].

**Theorem 6.2** (Excess risk of $\mathcal{A}_{\mathsf{NSGD}}$). *In Algorithm 4, let $\eta = R/\left(L \cdot n \cdot \max\left(\sqrt{n}, \frac{\sqrt{d\,\log(1/\beta)}}{\alpha}\right)\right)$. Then, for any $\theta \in (6\exp(-n/2), 1)$, with probability at least $1 - \theta$ over the randomness in both the sample and the algorithm, we have*

$$\varepsilon_{\mathsf{risk}}(\mathcal{A}_{\mathsf{NSGD}}) = RL \cdot O\left(\max\left(\frac{\log(n)\log(n/\theta)}{\sqrt{n}}, \frac{\sqrt{d\,\log(1/\beta)}}{\alpha\,n}\right)\right)$$

---

**Algorithm 4** $\mathcal{A}_{\mathsf{NSGD}}$: Noisy SGD for convex losses

---

**Require:** Private dataset $S = (z_1, \ldots, z_n) \in \mathcal{Z}^n$, step size $\eta$; privacy parameters $\alpha \leq 1$, $\beta \ll 1/n$

1: Set noise variance $\sigma^2 := \frac{8\,L^2\,\log(1/\beta)}{\alpha^2}$
2: Choose an arbitrary initial point $x^1 \in \mathcal{X}$
3: **for** $t = 1$ to $n^2 - 1$ **do**
4:     Sample $\mathbf{i}_t \sim \mathsf{Unif}([n])$
5:     $x^{t+1} := \mathsf{Proj}_{\mathcal{X}}\left(x^t - \eta \cdot \left(\nabla \ell(x^t, z_{\mathbf{i}_t}) + \mathbf{G}_t\right)\right)$, where $\mathbf{G}_t \sim \mathcal{N}\left(\mathbf{0}, \sigma^2 \mathbb{I}_d\right)$ drawn independently each iteration
6: **return** $\overline{x} = \frac{1}{n^2} \sum_{t=1}^{n^2} x^t$

---

*Proof.* Fix any confidence parameter $\theta \geq 6\exp(-n/2)$. First, for any data set $S \in \mathcal{Z}^n$ and any step size $\eta > 0$, by Lemma B.1 in Appendix B, we have the following high-probability guarantee on the training error of $\mathcal{A}_{\mathsf{NSGD}}$:
With probability at least $1 - \theta/3$, we have

$$\varepsilon_{\mathsf{opt}}(\mathcal{A}_{\mathsf{NSGD}}) \triangleq F_S(\overline{x}) - \min_{x \in \mathcal{X}} F_S(x) \leq \frac{R^2}{\eta\,n^2} + 7RL\frac{\sqrt{\log(1/\beta)\log(12/\theta)}}{\alpha n} + \eta L^2\left(32\frac{d\,\log(1/\beta)}{\alpha^2} + 1\right)$$

where the probability is over the sampling in step 4 and the independent Gaussian noise vectors $\mathbf{G}_1, \ldots, \mathbf{G}_{n^2}$. Given the setting of $\eta$ in the theorem, we get

$$\varepsilon_{\mathsf{opt}}(\mathcal{A}_{\mathsf{NSGD}}) \leq 8RL\max\left(\frac{1}{\sqrt{n}}, \frac{\sqrt{d\,\log(1/\beta)}}{\alpha\,n}\right) + 33RL\,\frac{d\frac{\log(1/\beta)}{\alpha^2}}{n \cdot \max\left(\sqrt{n}, \frac{\sqrt{d\,\log(1/\beta)}}{\alpha}\right)}$$

$$\leq 8RL\max\left(\frac{1}{\sqrt{n}}, \frac{\sqrt{d\,\log(1/\beta)}}{\alpha\,n}\right) + 33RL\,\frac{\sqrt{d\,\log(1/\beta)}}{n\,\alpha}$$

$$= RL \cdot O\left(\max\left(\frac{1}{\sqrt{n}}, \frac{\sqrt{d\,\log(1/\beta)}}{n\,\alpha}\right)\right). \tag{7}$$

Next, it is not hard to show that $\mathcal{A}_{\mathsf{NSGD}}$ attains the same UAS bound as $\mathcal{A}_{\mathsf{rSGD}}$ (Theorem 3.3). Indeed, the only difference is the noise addition in gradient step; however, this does not impact the stability analysis. This is because the sequence of noise vectors $\{\mathbf{G}_1, \ldots, \mathbf{G}_{n^2}\}$ is the same for the trajectories corresponding to the pair $S$, $S'$ of neighboring datasets. Hence, the argument basically follows the same lines of the proof of Theorem 3.3 since the noise terms cancel out. Thus, we conclude that for any pair $S \simeq S'$ of neighboring datasets, with probability at least $1 - \exp(n/2) \geq 1 - \theta/6$ (over the randomness of $\mathcal{A}_{\mathsf{NSGD}}$), the uniform argument stability of $\mathcal{A}_{\mathsf{NSGD}}$ is bounded as: $\delta_{\mathcal{A}_{\mathsf{NSGD}}} \leq 4L\eta\left(\sqrt{T} + \frac{T}{n}\right)$, where $T = n^2$. Given the setting of $\eta$ in the theorem, this bound reduces to $8R/\max\left(\sqrt{n}, \frac{\sqrt{d\,\log(1/\beta)}}{\alpha}\right)$.

Hence, by Theorem 2.2, with probability at least $1 - \theta/3$ (over the randomness in both the i.i.d. dataset $S$ and the algorithm), the generalization error of $\mathcal{A}_{\mathsf{NSGD}}$ is bounded as

$$\varepsilon_{\mathsf{gen}}(\mathcal{A}_{\mathsf{NSGD}}) \leq \frac{8c\,RL\,\log(n)\log(6n/\theta)}{\max\left(\sqrt{n}, \frac{\sqrt{d\,\log(1/\beta)}}{\alpha}\right)} + \frac{c\,\sqrt{\log(6/\theta)}}{\sqrt{n}} = RL \cdot O\left(\frac{\log(n)\log(n/\theta)}{\sqrt{n}}\right), \tag{8}$$

where $c$ in the first bound is a universal constant.

Now, using (7), (8), and Lemma 2.1, we finally conlcude that with probability at least $1 - \theta$ (over randomness in the sample $S$ and the internal randomness of $\mathcal{A}_{\mathsf{NSGD}}$), the excess population risk of $\mathcal{A}_{\mathsf{NSGD}}$ is bounded as

$$\varepsilon_{\mathsf{risk}}(\mathcal{A}_{\mathsf{NSGD}}) \leq \varepsilon_{\mathsf{opt}}(\mathcal{A}_{\mathsf{NSGD}}) + \varepsilon_{\mathsf{gen}}(\mathcal{A}_{\mathsf{NSGD}}) + \varepsilon_{\mathsf{approx}}$$

$$= RL \cdot O\left( \max\left( \frac{1}{\sqrt{n}}, \frac{\sqrt{d \, \log(1/\beta)}}{\alpha \, n} \right) + \frac{\log(n) \log(n/\theta)}{\sqrt{n}} + \frac{\sqrt{\log(1/\theta)}}{\sqrt{n}} \right)$$

$$= RL \cdot O\left( \max\left( \frac{\log(n) \log(n/\theta)}{\sqrt{n}}, \frac{\sqrt{d \, \log(1/\beta)}}{\alpha \, n} \right) \right),$$

which completes the proof. $\qquad\qquad\qquad\qquad\qquad\qquad\qquad\qquad\qquad\qquad\qquad\qquad\qquad\qquad\quad\square$

**Remark 6.3.** *Using the expectation guarantee on UAS given in Theorem 3.3 and following similar steps of the analysis above, we can also show that the expected excess population risk of $\mathcal{A}_{\mathsf{NSGD}}$ is bounded as:*

$$\mathbb{E}\left[\varepsilon_{\mathsf{risk}}\left(\mathcal{A}_{\mathsf{NSGD}}\right)\right] = RL \cdot O\left( \max\left( \frac{1}{\sqrt{n}}, \frac{\sqrt{d \, \log(1/\beta)}}{\alpha \, n} \right) \right).$$

## 6.2 Nonsmooth Stochastic Convex Optimization with Multi-pass SGD

Another application of our results concerns obtaining optimal excess risk for stochastic nonsmooth convex optimization via multi-pass SGD. It is known that one-pass SGD is guaranteed to have optimal excess risk, which can be shown via martingale arguments that trace back to the stochastic approximation literature [37, 25].

Using our UAS bound, we show that Algorithms 2 and 3 can recover nearly-optimal high-probability excess risk bounds by making $n$ passes over the data. Analogous bounds hold for Algorithm 1, however these are less interesting from a computational efficiency perspective.

### 6.2.1 Risk Bounds for Sampling-with-Replacement Stochastic Gradient Descent

**Theorem 6.4.** *Let $\mathcal{X} \subseteq \mathcal{B}(0, R)$ and $\mathcal{F} = \mathcal{F}_{\mathcal{X}}^0(L)$. The sampling with replacement stochastic gradient descent (Algorithm 2) with $T = n^2$ iterations and $\eta = \frac{R}{L \, n^{3/2}}$ satisfies for any $12 \exp\{-n^2/32\} < \theta < 1$.*

$$\mathbb{P}\left[ \varepsilon_{risk}(\mathcal{A}_{\mathsf{rSGD}}) = O\left( \frac{cLR}{\sqrt{n}} \log(n) \log(\frac{n}{\theta}) \right) \right] \leq \theta.$$

It should be noted that, similarly to Remark 6.3, if we are only interested in expectation risk bounds, one can shave off the polylogarithmic factor above, which is optimal for the expected excess risk.

*Proof.* Let $\mathbf{S} \sim \mathcal{D}^n$ an i.i.d. random sample for the stochastic convex program, and apply on these data the algorithm $\mathcal{A}_{\mathsf{rSGD}}$ for constant step size $\eta > 0$ and $T$ iterations.

We consider $\theta > 0$ such that $\theta > 12 \exp\{-T/32\}$. Notice that the sampling-with-replacement stochastic gradient is a bounded first-order stochastic oracle for the empirical objective. It is direct to verify that the assumptions of Lemma B.1 are satisfied with $\sigma = 0$. Hence, by Lemma B.1, we have that, with probability at least $1 - \theta/3$

$$\varepsilon_{\mathsf{opt}}(\mathcal{A}_{\mathsf{rSGD}}) \leq O\left( LR\sqrt{\frac{2 \log(12/\theta)}{T}} + \frac{R^2}{\eta T} + \eta L^2 \right).$$

On the other hand, Theorem 3.3 together with Theorem 2.2, guarantees that with probability at least $1 - \theta/3$, we have
$$|\varepsilon_{\text{gen}}(\mathcal{A}_{\text{rSGD}})| \leq O\Big(L^2\big[\sqrt{T}\eta + \frac{T\eta}{n}\big]\log n \log(6n/\theta) + LR\sqrt{\log(6/\theta)}n\Big).$$

Finally, Lemma 2.1 ensures that with probability $1 - \theta/3$
$$\varepsilon_{\text{approx}} \leq LR\sqrt{\frac{2\log(3/\theta)}{n}}.$$

By the union bound and the excess risk decomposition (3), we have that, with probability $1 - \theta$,
$$
\begin{aligned}
\varepsilon_{\text{risk}}(\mathcal{A}) &= O\Big(LR\sqrt{\frac{\log(1/\theta)}{T}} + \frac{R^2}{\eta T} + \eta L^2 + L^2\eta\big(\sqrt{T} + \frac{T}{n}\big)\log(n)\log(\frac{6n}{\theta}) \\
&\quad + LR\sqrt{\frac{\log(6/\theta)}{n}} + LR\sqrt{\frac{\log(3/\theta)}{n}}\Big) \\
&= O\Big(\frac{LR}{\sqrt{n}}\log(n)\log(\frac{n}{\theta})\Big),
\end{aligned}
$$

where only at the last step we replaced by the choice of step size and number of iterations from the statement. $\qquad\square$

### 6.2.2 Risk Bounds for Fixed-Permutation Stochastic Gradient Descent

As a final application we provide a population risk bound based on the UAS of Algorithm 3. Similarly as in the case of sampling-with-replacement SGD, we need an optimization error analysis, which for completeness is provided in Appendix C, and it is based on the analysis of the incremental subgradient method [33].

Interestingly, the combination of the incremental method analysis for arbitrary permutation [33] and our novel stability bounds that also work for arbitrary permutation, guarantees generalization bounds for fixed permutation SGD without the need of reshuffling, or even any form of randomization. We believe this could be of independent interest.

**Theorem 6.5.** *Algorithm 3 with constant step size $\eta_k \equiv \eta = R/[Ln\sqrt{K}]$ and $K = n$ epochs is such that for every $0 < \theta < 1$,*
$$\mathbb{P}\Big[\varepsilon_{risk}(\mathcal{A}_{\text{PerSGD}}) > \frac{cLR}{\sqrt{n}}\log n \log(\frac{n}{\theta})\Big] \leq \theta,$$
*where $c > 0$ is an absolute constant.*

Similarly to the previous result, we can remove the polylogarithmic factor if we are only interested in expected excess risk guarantees.

*Proof.* By Corollary C.2
$$\varepsilon_{\text{opt}}(\mathcal{A}_{\text{PerSGD}}) \leq \frac{R^2}{nK\eta} + \frac{L^2(n+2)\eta}{2} = O\Big(\frac{LR}{\sqrt{n}}\Big),$$

by our choice of $K, \eta$. On the other hand, Theorem 3.4 guarantees the algorithm is $\delta$-UAS with probability 1, where $\delta = O(R/\sqrt{n})$. Therefore, by Theorem 2.2, we have that w.p. $1 - \theta/2$
$$|\varepsilon_{\text{gen}}(\mathcal{A}_{\text{PerSGD}})| \leq c\Big(\frac{LR}{\sqrt{n}}\log n \log(2n/\theta) + LR\sqrt{\frac{\log 2/\theta}{n}}\Big).$$

Finally, Lemma 2.1 ensures that with probability $1 - \theta/2$

$$\varepsilon_{\text{approx}} \leq LR\sqrt{\frac{2\log(2/\theta)}{n}}.$$

By the union bound and the excess risk decomposition (3), we have that, with probability at least $1 - \theta$,

$$
\begin{aligned}
\varepsilon_{\text{risk}}(\mathcal{A}_{\text{PerSGD}}) &\leq \varepsilon_{\text{opt}}(\mathcal{A}_{\text{PerSGD}}) + \varepsilon_{\text{gen}}(\mathcal{A}_{\text{PerSGD}}) + \varepsilon_{\text{approx}} \\
&= O\Big(\frac{LR}{\sqrt{n}} + \frac{LR}{\sqrt{n}}\log n\log(n/\theta) + LR\sqrt{\frac{\log 1/\theta}{n}} + LR\sqrt{\frac{\log(2/\theta)}{n}}\Big) \\
&= O\Big(\frac{LR}{\sqrt{n}}\log n\log(\frac{n}{\theta})\Big).
\end{aligned}
$$

$\square$

## 7 Discussion and Open Problems

In this work we provide sharp upper and lower bounds on uniform argument stability for first-order methods in stochastic nonsmooth convex optimization. Our lower bounds show inherent limitations of stability bounds compared to the smooth convex case, however we can still derive optimal population risk bounds by reducing the step size and running the algorithms for longer number of iterations. We provide applications of this idea for differentially-private noisy SGD, and for two versions of SGD (sampling-with-replacement and fixed-permutation SGD).

The first open problem regards lower bounds that are robust to general forms of algorithmic randomization. Unfortunately, the methods presented here are not robust in this respect, since random initialization would prevent the trajectories reaching the region of highly nonsmooth behavior of the objective (or doing it in such a way that it does not increase UAS). One may try to strengthen the lower bound by using a random rotation of the objective; however, this leads to an uninformative lower bound. Finding distributional constructions for lower bounds against randomization is a very interesting future direction.

Our privacy application provides optimal risk for an algorithm that runs for $n^2$ steps, which is impractical for large datasets. Other algorithms, e.g. in [17], run into similar limitations. Proving that quadratic running time is necessary for DP-SCO is a very interesting open problem that can be formalized in terms of the oracle complexity of stochastic convex optimization [34] under stability and/or privacy constraints.

## Footnotes

[1]In fact, for both GD and fixed-permutation SGD we can obtain w.p. 1 upper bounds on $\delta_{\mathcal{A}}(S, S')$, whereas for sampling-with-replacement SGD, we obtain a high-probability upper bound.

[2]For equivalence to hold it is necessary that the function is well-defined and satisfies (2) over an open set containing $\mathcal{X}$, see Thm. 3.61 in [4]. We will assume this is the case, which can be done w.l.o.g..

[3] Here, we are applying a bound for (scaled) Bernoulli rvs where the exponent is expressed in terms of the variance.

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

# A  Upper bounds on UAS of SGD when $T \leq n$

**Theorem A.1.** *Let $\mathcal{X} \subseteq \mathcal{B}(0, R)$ and $\mathcal{F} = \mathcal{F}_{\mathcal{X}}^0(L)$. Suppose $T \leq n$. The UAS of sampling-with-replacement stochastic gradient descent (Algorithm 2) satisfies uniform argument stability*

$$\mathbb{E}\left[\delta_{\mathcal{A}_{\mathrm{rSGD}}}\right] \leq \min\left(2R, \ 3L \frac{T-1}{n}\left(\sqrt{\sum_{t=1}^{T-1} \eta_t^2} + \frac{1}{n}\sum_{t=1}^{T-1}\eta_t\right)\right).$$

*Proof.* The bound of $2R$ is obtained directly from the diameter bound on $\mathcal{X}$. Therefore, we focus exclusively on the second term. Let $S \simeq S'$, and let $k \in [n]$ be the entry where both datasets differ. Let $(x^t)_{t\in[T]}, (y^t)_{t\in[T]}$ be the trajectories of Algorithm 2 on $S$ and $S'$, respectively, with $x^1 = y^1$.

Let $B_t$ denote the event that $\mathbf{i}_j = k$ for some $j \leq t$; that is, $B_t$ is the event that the index $k$ is sampled at least once in the first $t$ iterations. We note that

$$\mathbb{P}[B_t] \ = \ \frac{1}{n}\sum_{j=0}^{t-1}\left(1 - \frac{1}{n}\right)^j = 1 - \left(1 - \frac{1}{n}\right)^t \leq \frac{t}{n}.$$

Hence, we have

$$\mathbb{E}\left[\delta_T\right] \leq \frac{T-1}{n} \cdot \mathbb{E}\left[\delta_T \mid B_{T-1}\right]. \tag{9}$$

For the rest of the proof we bound $\mathbb{E}\left[\delta_T | B_{T-1}\right]$. To do this, we derive a recurrence for $\mathbb{E}\left[\delta_{t+1} | B_t\right]$. Note that $B_t$ is the union of two mutually exclusive events: $\{\mathbf{i}_t = k\} \cap \overline{B_{t-1}}$ and $B_{t-1}$, where $\overline{B_{t-1}}$ is the complement of $B_{t-1}$, i.e., the event "index $k$ is never sampled in the first $t$ iterations." Hence,

$$
\begin{aligned}
\mathbb{E}\left[\delta_{t+1}^2 \mid B_t\right] &= \mathbb{P}\left[\mathbf{i}_t = k, \overline{B_{t-1}} \mid B_t\right] \mathbb{E}\left[\delta_{t+1}^2 \mid \mathbf{i}_t = k, \overline{B_{t-1}}\right] + \mathbb{P}\left[B_{t-1} \mid B_t\right] \mathbb{E}\left[\delta_{t+1}^2 \mid B_{t-1}\right] \\
&\leq \mathbb{E}\left[\delta_{t+1}^2 \mid \mathbf{i}_t = k, \overline{B_{t-1}}\right] + \mathbb{E}\left[\delta_{t+1}^2 \mid B_{t-1}\right].
\end{aligned}
\tag{10}
$$

Now, conditioned on the past sampled coordinates $\mathbf{i}_1, \ldots, \mathbf{i}_{t-1}$, we have

$$
\begin{aligned}
\delta_{t+1} &= \|\mathsf{Proj}_{\mathcal{X}}[x^t - \eta_t \nabla f(x^t, z_{\mathbf{i}_t})] - \mathsf{Proj}_{\mathcal{X}}[y^t - \eta_t \nabla f(y^t, z'_{\mathbf{i}_t})]\| \\
&\leq \|x^t - y^t - \eta_t[\nabla f(x^t, z_{\mathbf{i}_t}) - \nabla f(y^t, z'_{\mathbf{i}_t})]\| \\
&\leq \mathbf{1}_{\{\mathbf{i}_t = k\}}(\delta_t + 2L\eta_t) + \mathbf{1}_{\{\mathbf{i}_t \neq k\}}\sqrt{\delta_t^2 + 4L^2\eta_t^2},
\end{aligned}
$$

where the last inequality is obtained from convexity and Lipschitzness of the objective. Now, squaring we get

$$
\begin{aligned}
\delta_{t+1}^2 &\leq \mathbf{1}_{\{\mathbf{i}_t = k\}}(\delta_t + 2L\eta_t)^2 + \mathbf{1}_{\{\mathbf{i}_t \neq k\}}(\delta_t^2 + 4L^2\eta_t^2) \\
&= \delta_t^2 + 4\eta_t^2 L^2 + \mathbf{1}_{\{\mathbf{i}_t = k\}}4L\eta_t\delta_t.
\end{aligned}
$$

From this formula we derive bounds for the two conditional expectations:

$$\mathbb{E}\left[\delta_{t+1}^2 \mid B_{t-1}\right] \leq \mathbb{E}\left[\delta_t^2 \mid B_{t-1}\right] + 4L^2\eta_t^2 + \frac{4L}{n}\eta_t\mathbb{E}\left[\delta_t \mid B_{t-1}\right] \tag{11}$$

$$\mathbb{E}\left[\delta_{t+1}^2 \mid \mathbf{i}_t = k, \overline{B_{t-1}}\right] \leq 4L^2\eta_t^2, \tag{12}$$

where (11) holds by independence of $\mathbf{i}_t$ and $B_{t-1}$, and in (12) we used that $\delta_t = 0$ conditioned on $\overline{B_{t-1}}$.

Combining (11) and (12) in (10), we get

$$\mathbb{E}\left[\delta_{t+1}^2 \mid B_t\right] \leq \mathbb{E}\left[\delta_t^2 \mid B_{t-1}\right] + 8L^2\eta_t^2 + \frac{4L}{n}\eta_t\mathbb{E}\left[\delta_t \mid B_{t-1}\right]$$

$$\mathbb{E}[\delta_T^2 \mid B_{T-1}] \leq 8L^2\sum_{t=1}^{T-1}\eta_t^2 + \frac{4L}{n}\sum_{t=1}^{T-1}\eta_t\mathbb{E}[\delta_t \mid B_{t-1}].$$

With this last bound we can proceed inductively to show that

$$\mathbb{E}\left[\delta_T \mid B_{T-1}\right] \leq \sqrt{8}L\sqrt{\sum_{t=1}^{T-1}\eta_t^2 + \frac{2L}{n}\sum_{t=1}^{T-1}\eta_t}.$$

The base case, $T = 0$, is evident, and the inductive step can be considered in two separate cases; namely, the case where $\mathbb{E}[\delta_T \mid B_{T-1}] \leq \max_{t \in [T-1]} \mathbb{E}[\delta_t \mid B_{t-1}]$, which can be obtained by the induction hypothesis; and the case where $\mathbb{E}[\delta_T \mid B_{T-1}] > \max_{t \in [T-1]} \mathbb{E}[\delta_t \mid B_{t-1}]$, for which

$$\mathbb{E}[\delta_T^2 | B_{T-1}] \leq 8L^2\sum_{t=1}^{T-1}\eta_t^2 + \frac{4L}{n}\sum_{t=1}^{T-1}\eta_t\mathbb{E}[\delta_t | B_{t-1}] \leq 8L^2\sum_{t=1}^{T-1}\eta_t^2 + \frac{4L}{n}\sum_{t=1}^{T-1}\eta_t\mathbb{E}[\delta_T | B_{T-1}].$$

Then

$$\mathbb{E}_{\mathbf{i}}\Big[\Big(\delta_T - \frac{2L}{n}\sum_{t=1}^{T-1}\eta_t\Big)^2 \,\Big|\, B_{T-1}\Big] \;\leq\; 8L^2\sum_{t=1}^{T-1}\eta_t^2 + \Big(\frac{2L}{n}\sum_{t=1}^{T-1}\eta_t\Big)^2,$$

and by the Jensen inequality

$$\mathbb{E}_{\mathbf{i}}\Big[\delta_T - \frac{2L}{n}\sum_{t=1}^{T-1}\eta_t\,\Big|\,B_{T-1}\Big] \;\leq\; \sqrt{8L^2\sum_{t=1}^{T-1}\eta_t^2 + \Big(\frac{2L}{n}\sum_{t=1}^{T-1}\eta_t\Big)^2} \;\leq\; \sqrt{8}L\sqrt{\sum_{t=1}^{T-1}\eta_t^2} + \frac{2L}{n}\sum_{t=1}^{T-1}\eta_t,$$

proving the inductive step. Finally, putting this together with (9) completes the proof. $\qquad\square$

**Theorem A.2.** *Let $\mathcal{X} \subseteq \mathcal{B}(0, R)$, $\mathcal{F} = \mathcal{F}^0_{\mathcal{X}}(L)$, $\pi$ be a uniformly random permutation over $[n]$, and $(\eta_t)_{t\in[T]}$ be a non-increasing sequence. Let $T \leq n$. The UAS of fixed-permutation stochastic gradient descent (Algorithm 3) satisfies uniform argument stability $\mathbb{E}\left[\delta_{\mathcal{A}_{\mathsf{PerSGD}}}\right] \leq \min\{2R, \sqrt{2}L\frac{T-1}{n}\sqrt{\sum_{t=1}^{T-1}\eta_t^2}\}$.*

**Observation A.3.** *Note that the bound above for fixed permutation SGD in Theorem A.2 is of the same order as that of sampling with replacement SGD in Theorem A.1. This is because $\sqrt{\sum_{t=1}^{T-1}\eta_t^2} \geq \frac{1}{\sqrt{T}}\sum_{t=1}^{T-1}\eta_t$ (by Cauchy-Schwarz inequality), and hence, when $T \leq n$, we would have $\sqrt{\sum_{t=1}^{T-1}\eta_t^2} \geq \frac{1}{\sqrt{n}}\sum_{t=1}^{T-1}\eta_t \geq \frac{1}{n}\sum_{t=1}^{T-1}\eta_t$.*

*Proof.* The stability bound of $2R$ is implied directly by the diameter of the feasible set. Let $S \simeq S'$, and let $(x^t)_{t\in[T]}, (y^t)_{t\in[T]}$ be the trajectories of Algorithm 3 on $S$ and $S'$, respectively, with $x^1 = y^1$.

Notice that since $\pi$ is a random permutation, we may assume w.l.o.g. that $\pi$ is the identity, whereas the perturbed coordinate between $S, S'$ is $\mathbf{i} \sim \mathsf{Unif}([n])$. The rest of the proof is a stability analysis conditioned on $\pi$ (which fixes all the randomness of the algorithm), but from the observation above this is the same as conditioning on the random perturbed coordinate $\mathbf{i}$.

Let $T \leq n$, and $\delta_t = \|x^t - y^t\|$ so that $\delta_1 = 0$. Conditioned on $\mathbf{i} = i$, we have that for all $t \leq T$,

$$\delta_{t+1}^2 \leq \begin{cases} 0 & t < i \\ 4\eta_t^2 L^2 & t = i \\ \delta_t^2 + 4\eta_t^2 L^2 & i < t \leq T \end{cases}$$

Indeed, for all $t \leq i$, $\delta_t = 0$. For $t = i$, we have

$$\begin{aligned}
\delta_{i+1} &= \|\mathsf{Proj}_{\mathcal{X}}[x^i - \eta_i\nabla f(x^i, z_i)] - \mathsf{Proj}_{\mathcal{X}}[y^i - \eta_i\nabla f(y^i, z_i')]\| \\
&\leq \|x^i - y^i - \eta_i[\nabla f(x^i, z_i) - \nabla f(y^i, z_i')]\| \\
&\leq 2L\eta_i,
\end{aligned}$$

where we used $x^i = y^i$, and that both gradients are bounded in norm by $L$. Finally, when $t > i$, we have $z_t = z_t'$, and therefore we can leverage the monotonicity of the subgradients

$$\begin{aligned}
\delta_{t+1}^2 &= \|\mathsf{Proj}_{\mathcal{X}}[x^t - \eta_t\nabla f(x^t, z_t)] - \mathsf{Proj}_{\mathcal{X}}[y^t - \eta_t\nabla f(y^t, z_t)]\|^2 \\
&\leq \delta_t^2 + 4L^2\eta_t^2 - 2\eta_t\langle\nabla f(x^t, z_t) - \nabla f(y^t, z_t), x^t - y^t\rangle \\
&\leq \delta_t^2 + 4L^2\eta_t^2.
\end{aligned}$$

Unravelling this recursion, we get $\mathbb{E}[\delta_{t+1}^2|\mathbf{i} = i] \leq 4L^2 \sum_{s=i}^{t} \eta_s$, so by the law of total expectation:

$$\mathbb{E}[\delta_t] = \frac{1}{n}\sum_{i=1}^{n}\mathbb{E}[\delta_t|\mathbf{i} = i] \leq \frac{1}{n}\sum_{i=1}^{n}\sqrt{\mathbb{E}[\delta_t^2|\mathbf{i} = i]} \leq \frac{2L}{n}\sum_{i=1}^{t-1}\sqrt{\sum_{s=i}^{t-1}\eta_s^2}$$

$$\leq \frac{2L}{n}\sum_{i=1}^{t-1}\sqrt{(t-i)}\eta_i \leq \frac{2L}{n}\sqrt{\left(\sum_{i=1}^{t-1}(t-i)\right)\left(\sum_{i=1}^{t-1}\eta_i^2\right)}$$

$$\leq \frac{2L}{n}\sqrt{\frac{(t-1)^2}{2}\sum_{i=1}^{t-1}\eta_i^2} = \frac{\sqrt{2}L(t-1)}{n}\sqrt{\sum_{i=1}^{t-1}\eta_i^2}.$$

where the first inequality holds by the Jensen inequality, the second inequality comes from the bound on the conditional expectation, the third inequality from the non-increasing stepsize assumption, and the fourth inequality is from Cauchy-Schwarz. Since averaging can only improve stability, we conclude the result. $\square$

# B  High-probability Bound on Optimization Error of SGD with Noisy Gradient Oracle

It is known that standard online-to-batch conversion technique can be used to provide high-probability bound on the optimization error (i.e., the excess empirical risk) of stochastic gradient descent. For the sake of completeness and to make the paper more self-contained, we re-expose this technique here for stochastic gradient descent with noisy gradient oracle. This is done in the following lemma, which is used in the proofs of our results in Section 6.

**Lemma B.1** (Optimization error of SGD with noisy gradient oracle). *Let $S = (z_1, \ldots, z_n) \in \mathcal{Z}^n$ be a dataset. Let $F_S(x) = \frac{1}{n}\sum_{i\in[n]} f(x, z_i)$ be the empirical risk associated with S, where for every $z \in \mathcal{Z}$, $f(\cdot, z)$ is convex, L-Lipschitz function over $\mathcal{X} \subseteq \mathcal{B}(0, R)$ for some $L, R > 0$. Consider the stochastic (sub)gradient method:*

$$x^{t+1} = x^t - \eta \cdot \mathbf{g}(x, \xi_t) \qquad (\forall t = 0, \ldots, T-1),$$

*with output $\overline{x}^T = \frac{1}{T}\sum_{t\in[T]} x^t$; where $\xi_1, \ldots, \xi_T$ are drawn uniformly from from S with replacement, and for all $z \in \mathcal{Z}$, $\mathbf{g}(., z) : \mathcal{X} \to \mathbb{R}^d$ is a random map (referred to as noisy gradient oracle) that satisfies the following conditions:*

1. *Unbiasedness: For every $x \in \mathcal{X}, z \in \mathcal{Z}$, $\mathbb{E}[\mathbf{g}(x, z)] = \nabla f(x, z)$, where the expectation is taken over the randomness in the gradient oracle $\mathbf{g}(\cdot, z)$.*

2. *Sub-Gaussian gradient noise: There is $\sigma^2 \geq 0$ such that for every $x \in \mathcal{X}, z \in \mathcal{Z}$, $\mathbf{g}(x, z) - \nabla f(x, z)$ is $\sigma^2$-sub-Gaussian random vector; that is, for every $x \in \mathcal{X}, z \in \mathcal{Z}$, $\langle \mathbf{g}(x, z) - \nabla f(x, z), u \rangle$ is $\sigma^2$-sub-Gaussian random variable $\forall u \in \mathcal{B}(0, 1)$.*

3. *Independence of the gradient noise across iterations: conditioned on any fixed realization of $(\xi_t)_{t\in[T]}$ the sequence of random maps $\mathbf{g}(\cdot, \xi_1), \ldots, \mathbf{g}(\cdot, \xi_T)$ is independent. (Here, randomness comes only from the gradient oracle.)*

*Then, for any $\theta \in (4e^{-T/32}, 1)$, with probability at least $1 - \theta$, the optimization error (i.e., the excess empirical risk) of this method is bounded as*

$$\varepsilon_{\text{opt}} \leq (LR + \sigma(R + \eta L)) \sqrt{\frac{2\log(4/\theta)}{T}} + \frac{R^2}{2\eta T} + \eta\left(\frac{L^2}{2} + d\sigma^2\right).$$

*Proof.* Let $x_S^* \in \arg\min\limits_{x \in \mathcal{X}} F_S(x)$. By convexity of the empirical loss, we have

$$F_S(\overline{x}^T) - F_S(x_S^*) \leq \frac{1}{T} \sum_{t \in [T]} F_S(x^t) - F_S(x_S^*)$$

$$= \frac{1}{T} \sum_{t \in [T]} [F_S(x^t) - f(x^t, \xi_t)] + \frac{1}{T} \sum_{t \in [T]} [f(x_S^*, \xi_t) - F_S(x_S^*)] + \frac{1}{T} \sum_{t \in [T]} [f(x^t, \xi_t) - f(x_S^*, \xi_t)]. \quad (13)$$

Since $(\xi_t)_{t \in [T]}$ are sampled uniformly without replacement from $S$, we have

$$\mathop{\mathbb{E}}_{\xi_t \mid x^1, \ldots, x^{t-1}} \left[ f(x^t, \xi_t) \mid x^1, \ldots, x^{t-1}, x^t = v \right] = F_S(v),$$

for all $v \in \mathcal{X}, t \in [T]$. Moreover, since the range of $f$ lies in $[-LR, LR]$, it follows that $Y_t := \sum_{j=1}^{t} f(x^j, \xi_j)$, $t \in [T]$ is a martingale with bounded differences (namely, bounded by $2LR$). Therefore, by Azuma's inequality, the first term in (13) satisfies

$$\mathbb{P}\left[ \frac{1}{T} \sum_{t \in [T]} [F_S(x^t) - f(x^t, \xi_t)] > LR\sqrt{\frac{2\log\frac{4}{\theta}}{T}} \right] \leq \frac{\theta}{4}. \quad (14)$$

By Hoeffding's inequality, the second term in (13) also satisfies the same bound; namely,

$$\mathbb{P}\left[ \frac{1}{T} \sum_{t \in [T]} [f(x_S^*, \xi_t) - F_S(x_S^*)] > LR\sqrt{\frac{2\log\frac{4}{\theta}}{T}} \right] \leq \frac{\theta}{4}. \quad (15)$$

Using similar analysis to that of the standard online gradient descent analysis [44], the last term in (13) can be bounded as

$$\frac{1}{T} \sum_{t \in [T]} [f(x^t, \xi_t) - f(x_S^*, \xi_t)] \leq \frac{R^2}{2T\eta} + \frac{1}{T} \sum_{t \in [T]} \langle \nabla f(x^t, \xi_t) - \mathbf{g}(x^t, \xi_t), x^t - x_S^* \rangle + \frac{\eta}{2T} \sum_{t \in [T]} \|\nabla \mathbf{g}(x^t, \xi_t)\|^2$$

$$= \frac{R^2}{2T\eta} + \frac{1}{T} \sum_{t \in [T]} \langle \nabla f(x^t, \xi_t) - \mathbf{g}(x^t, \xi_t), x^t - x_S^* - \eta \nabla f(x^t, \xi_t) \rangle + \frac{\eta}{2T} \sum_{t \in [T]} \|\mathbf{g}(x^t, \xi_t) - \nabla f(x^t, \xi_t)\|^2$$

$$+ \frac{\eta}{2T} \sum_{t \in [T]} \|\nabla f(x^t, \xi_t)\|^2$$

$$\leq \frac{R^2}{2T\eta} + \frac{\eta L^2}{2} + \frac{1}{T} \sum_{t \in [T]} \langle \nabla f(x^t, \xi_t) - \mathbf{g}(x^t, \xi_t), x^t - x_S^* - \eta \nabla f(x^t, \xi_t) \rangle + \frac{\eta}{2T} \sum_{t \in [T]} \|\mathbf{g}(x^t, \xi_t) - \nabla f(x^t, \xi_t)\|^2$$

$$\quad (16)$$

By the properties of the gradient oracle stated in the lemma, we can see that for any fixed realization of $(x^t, \xi_t)_{t \in [T]}$, the second term in (16) is $(2R + \eta L)^2 \frac{\sigma^2}{T}$-sub-Gaussian random variable. Hence,

$$\mathbb{P}\left[\frac{1}{T}\sum_{t \in [T]}\langle\nabla f(x^t, \xi_t) - \mathbf{g}(x^t, \xi_t),\ x^t - x_S^* - \eta\nabla f(x^t, \xi_t)\rangle > (2R + \eta L)\,\sigma\sqrt{\frac{2\log(4/\theta)}{T}}\right] \le \frac{\theta}{4}. \quad (17)$$

Let $U_t := \|\mathbf{g}(x^t, \xi_t) - \nabla f(x^t, \xi_t)\|^2$, $t \in [T]$. Note that $\mathbb{E}[U_t] \le d\sigma^2$. Moreover, observe (e.g., by [36, Lemma 1.12]) that for any fixed realization of $x^t, \xi_t$, $V_t := U_t - \mathbb{E}[U_t]$ is a sub-exponential random variable with parameter $16d\sigma^2$; namely, $\mathbb{E}[\exp(\lambda V_t)] \le \exp(128\lambda^2\sigma^4 d^2)$, $\lambda \le \frac{1}{16\sigma^2 d}$. Hence, by a standard concentration argument (e.g., Bernstein's inequality), we have

$$\mathbb{P}\left[\frac{\eta}{2T}\sum_{t \in [T]}\|\mathbf{g}(x^t, \xi_t) - \nabla f(x^t, \xi_t)\|^2 > \frac{\eta}{2}d\sigma^2 + 16\eta d\sigma^2\,\frac{\log(4/\theta)}{T}\right] \le \theta/4. \quad (18)$$

Putting (17) and (18) together, and noticing that $T > 32\log(4/\theta)$, we conclude that with probability at least $1 - \theta/2$, the third term of (13) is bounded as

$$\frac{1}{T}\sum_{t \in [T]}[f(x^t, \xi_t) - f(x_S^*, \xi_t)] \le \frac{R^2}{2T\eta} + \frac{\eta L^2}{2} + \eta\sigma^2 d + (2R + \eta L)\,\sigma\sqrt{\frac{2\log(4/\theta)}{T}}.$$

Hence, by the union bound, we conclude that with probability at least $1 - \theta$, the excess empirical risk of the stochastic subgradient method is bounded as

$$\varepsilon_{\mathrm{opt}} \le (LR + \sigma(2R + \eta L))\sqrt{\frac{2\log(4/\theta)}{T}} + \frac{R^2}{2\eta T} + \eta\,(\frac{L^2}{2} + d\sigma^2).$$

$\square$

## C  Empirical Risk of Fixed-Permutation SGD

Our optimization error analysis is based on [33, Lemma 2.1].

**Lemma C.1.** *Let us consider the fixed permutation stochastic gradient descent (Algorithm 3), for arbitrary permutation (i.e., not necessarily random) and with constant step size over each epoch (i.e., $\eta_{(k-1)n+t} \equiv \eta_k$ for all $t \in [n]$, $k \in [K]$). Then*

$$\eta_k[F_S(x^k) - F_S(y)] \le \frac{1}{2n}[\|x^k - y\|^2 - \|x^{k+1} - y\|^2] + \frac{\eta_k^2 L^2(n+2)}{2} \qquad (\forall y \in \mathcal{X}).$$

*Proof.* First, since the permutation is arbitrary, w.l.o.g. $\pi$ is the identity (we make this choice only for notational convenience). Let now $y \in \mathcal{X}$. At each round, the recursion of SGD implies that

$$
\begin{aligned}
\|x_{t+1}^k - y\|^2 &= \|\mathsf{Proj}_{\mathcal{X}}[x_t^k - \eta_k\nabla f(x_t^k, z_t)] - \mathsf{Proj}_{\mathcal{X}}(y)\|^2 \\
&\le \|x_t^k - \eta_k\nabla f(x_t^k, z_t) - y\|^2 \\
&= \|x_t^k - y\|^2 + \eta_k^2 L^2 - 2\eta_k\langle\nabla f(x_t^k, z_t), x_t^k - y\rangle \\
&\le \|x_t^k - y\|^2 + \eta_k^2 L^2 - 2\eta_k[f(x_t^k, z_t) - f(y, z_t)].
\end{aligned}
$$

Let $r_t := \|x^t - y\|$. Summing up these inequalities from $t = 1, \ldots, n$

$$
\begin{aligned}
r_{n+1}^2 - r_1^2 \;\; \leq \;\; & nL^2\eta_k^2 + 2\eta_k \sum_{t=1}^{n}[f(x^k, z_t) - f(x_t^k, z_t)] - 2\eta_k n[F_S(x^k) - F_S(y)] \\
\leq \;\; & nL^2\eta_k^2 + 2\eta_k \sum_{t=1}^{n} L\|x^k - x_t^k\| - 2\eta_k n[F_S(x^k) - F_S(y)] \\
\leq \;\; & nL^2\eta_k^2 + 2\eta_k^2 L^2 \sum_{t=1}^{n} t - 2\eta_k n[F_S(x^k) - F_S(y)] \\
= \;\; & \eta_k^2 L^2 n + \eta_k^2 L^2 n(n+1) - 2\eta_k n[F_S(x^k) - F_S(y)].
\end{aligned}
$$

Re-arranging terms we obtain the result. $\qquad\square$

Using the previous lemma, it is straightforward to derive the optimization accuracy of the method.

**Corollary C.2.** *The fixed permutation stochastic gradient descent (Algorithm 3), for arbitrary permutation (i.e., not necessarily random) and with constant step size over each epoch (i.e., $\eta_{(k-1)n+t} \equiv \eta_k$ for all $t \in [n]$, $k \in [K]$). satisfies*

$$
\varepsilon_{opt} \leq \frac{\|x^1 - x^*(S)\|^2}{2n \sum_k \eta_k} + \frac{L^2(n+2) \sum_k \eta_k^2}{2} \cdot \frac{1}{\sum_k \eta_k}.
$$

*Proof.* By convexity and Lemma C.1, we have

$$
\begin{aligned}
F_s(\overline{x}^K) - F_S(x^*(S)) \;\; \leq \;\; & \frac{1}{\sum_{k=1}^{K} \eta_k} \sum_{k=1}^{K} \eta_k[F_S(x^k) - f_S(x^*(S))] \\
\leq \;\; & \frac{1}{\sum_{k=1}^{K} \eta_k} \left[ \frac{1}{2n}\|x^1 - x^*(S)\|^2 + \frac{L^2(n+2)}{2} \sum_{k=1}^{K} \eta_k^2 \right],
\end{aligned}
$$

which proves the result. $\qquad\square$