[Reviews · NeurIPS 2020]

Review 1

Summary and Contributions: This paper provides upper and lower bounds on the stability of (S)GD when the loss is nonsmooth and convex. The techniques are novel and they use their results to analyze a new Differentially private algorithm for stochastic optimization. They also provide generalization bounds for multi-pass nonsmooth SGD.

Strengths: Overall a great paper. Very well written, and well organized. Results are definitely novel, sound and I am sure the community will receive this work well.

Weaknesses: It would be nice if you discussed in the Broader Impact section how your results may be used by others. It would also be nice to see future research directions. It looks like you already solved the problem in hand (judging from the lower bounds), if I am mistaken could you comment more about it?

Correctness: looked at some of the proofs in detail and the seem to be correct.

Clarity: Yes, very well written

Relation to Prior Work: yes

Reproducibility: Yes

Additional Feedback: I just have a few questions. Q: is it possible to prove a statement similar to Lemma 3.1 where instead of having x^1 = y^1 you have x^1 = y^1 + v where v is such that || v || <= some small number? Q: what could be improved if you were to assume that the loss functions are also strongly-convex? also, I don’t think the equality in 182 is correct (although it is an upper bound). UPDATE AFTER AUTHOR FEEDBACK: Thanks for the clarifications! I think this is good work so will leave my score unchanged.


Review 2

Summary and Contributions: The paper addresses the uniform stability of SGD on nonsmooth convex loss functions. To be specific, the author presents sharp bounds of L2 sensitivity (UAS) of SGD variants, therefore obtaining the optimal excess empirical risk. As a consequence, the paper provides the generalization bounds for multi-pass nonsmooth SGD and utility bounds for the differentially private variant.

Strengths: This is the first work to consider the stabillity of SGD for nonsmooth convex losses with well-established theories and specific consequences in applications.

Weaknesses: It seems that this work lacks the motivating examples in practice.The paper did not mention any specific example of nonsmooth convex losses. In machine learning, the nonsmoothness of losses often comes from regularization terms. In these cases, instead of finding subgradients, it is better to use proximal gradient methods that have faster convergence rates.

Correctness: All the claims and methods are clear and correct.

Clarity: Overall, it is a well written paper.

Relation to Prior Work: Yes

Reproducibility: Yes

Additional Feedback: 1. Section 1 and Section 2 are a liittle bit messy, and some sentences are redundant. It is better to reorganize these two sections in the next draft. 2. When writing subgradients, please use $\partial$ instead of $\nabla$ to avoid confusion.


Review 3

Summary and Contributions: The authors study the uniform stability of SGD with convex and non-smooth loss functions. Based on this, the authors develop both high-probability generalization bounds and generalization bounds in expectation for multi-pass SGD, which is dimension-free. The authors further apply this result to develop a differentially private stochastic optimization algorithm for non-smooth optimization problems. -------------- After Authors' response -------------- Thank you for your response. It is still not quite clear to me how the technique in [1] applies to the uniform sampling as considered in this paper. It would be helpful for the readers if the authors can provide details in the final version. I agreed that constants can be computed. It will be better to derive explicit privacy guarantees in the revised version. I would also like to see more discussions on the paper "Fine-grained analysis" since their work also establishes stability bounds for nonsmooth convex losses. Although they consider generally on-average stability bounds, as far as I can see their analysis for non-smooth loss applies to uniform stability. Overall, I like this paper. It has interesting results on differential privacy for learning with non-smooth loss functions. The lower bounds on uniform stability are also interesting.

Strengths: Previous stability analysis of SGD requires loss functions to be strongly smooth. An essential property in the smooth case is that the gradient update is non-expansive. This paper uses the monotonicity of the gradient to tackle the non-expansiveness and successfully imply interesting stability bounds. The authors further show that the derived stability bounds cannot be further improved by establishing matching lower bounds for some elegantly constructed loss functions. An application of this is to develop optimal generalization bounds $O(1/\sqrt{n})$ for multi-pass SGD by taking step size $\eta=(Tn)^{-1/2}$ and $T=n^2$. Based on the stability analysis, the authors further develop a computationally efficient and differentially private algorithm for non-smooth learning which enjoys the optimal generalization bounds. This algorithm is significantly more computationally efficient than [2], and improves the complexity of the algorithm in [16] by removing a logarithmic factor and is substantially simple.

Weaknesses: - Below eq (3), for the upper bound of $\delta_t$ the right-hand side should be $2\sum_s\eta_sa_s$ instead of $2\sum_s\eta_sa_s\delta_s$. - It is misleading to claim that it is the first work to address the stability of SGD for non-smooth convex loss functions as there are indeed existing work which already addressed stability of stochastic optimization with non-smooth loss. It would be interesting to add some discussions or comparison with these references mentioned below: 1. “Fine-Grained Analysis of Stability and Generalization for Stochastic Gradient Descent”. ICML 2020. In this paper, their work relaxes the smoothness to $\alpha$-Holder continuity of (sub)gradients, which include the non-smooth loss functions in this paper as $\alpha=0$. Their stability analysis also improves the optimal generalization bounds $O(1/\sqrt{n})$ for multi-pass SGD with $T=O(n^2)$. It seems to me that the main technical novelty appeared in the proof of Lemma 3 which studied \delta_t^2 (as opposed to the study of \delta_t in Hardt et al’s paper) using the approximate contraction for the gradient mapping for the non-smooth loss which has already explored in the above paper. Similar ideas have already explored in the above reference in a more general setting. 2. Private Stochastic Convex Optimization: Efficient Algorithms for Non-smooth Objectives, Arxiv preprint (2020). In this Arxiv preprint, the authors developed a different differentially private algorithm (Private FTRL) for non-smooth learning problems which can also achieve optimal generalization bounds. - The authors indicate that Theorem 5.1 on privacy guarantees follow from the same line of Theorem 2.1 in [3] but omit the proof. Furthermore, the authors mention that they replace the privacy analysis of the Gaussian mechanism with the tighter Moments Accountant method [1]. However, the analysis in [1] consider the Poisson sampling while Algorithm 2 considers uniform sampling with replacement. Furthermore, the moment bound in [1] is asymptotic. Therefore, it is not clear to me how to derive Theorem 5.1. I would recommend the authors to include the details for completeness as the differentially private SGD is an important application of the stability analysis for non-smooth loss functions.

Correctness: The claims and method are correct.

Clarity: The paper is very well written and is very clear to follow.

Relation to Prior Work: The difference of this work from previous contributions is clearly discussed.

Reproducibility: Yes

Additional Feedback:


Review 4

Summary and Contributions: This paper provides stability guarantees on SGD for solving nonsmooth Lipschitz convex functions. It allows the authors to further derive generalization error of multi-pass SGD and develop differentially private SGD.

Strengths: The paper is relavent to the Neurips community since it is both related to learning theory and optimization. The key contribution is a uniform stability upper bound on sgd for minimizing nonsmooth convex losses, which extends the previously known bound for smooth convex minimization problems, see e.g. Train faster, generalize better: Stability of stochastic gradient descent.

Weaknesses: Some minor comments: 1. The definition of $\varepsilon_{risk}$ is not consistent in the paper. See page 1 and page 4. 2. I'm not sure if I missed something. The paper adopts the gradient notation for subgradient. Since this paper primarily deals with nonsmooth functions, it might be better to use another notation. 3. In the remark after Theorem 4.1, the authors claim the optimality of multi-pass SGD for excess risk with properly chosen stepsize. I think this result might deserve a place in the main text. 4. Since the authors have obtained bound on sgd with fixed permutation, I'm curious to see what privacy guarantee one can say about sgd with fixed permutation. Even further, what can we say about SGD without replacement (i.e. sgd with different permutations in each epoch)?

Correctness: The claims (most importantly Lemma 3.1) seem correct. I checked part of the proof.

Clarity: The paper is well written and I enjoyed reading the paper.

Relation to Prior Work: Yes, the authors provide adequate account on related and prior work.

Reproducibility: Yes

Additional Feedback:

[Author Response · NeurIPS 2020]

We would like to thank the reviewers for their careful revision, positive comments and constructive criticism. We address below some of the major criticism raised by reviewers. Due to space limitations, not all suggestions can be included in the conference file, but we will make sure to include them in the full version of the paper.

- *Broader Impact:* 2 reviewers commented that the paper is short on the Broader Impact section. In the updated conference version we will add a more thorough discussion on private data analysis and its societal impact.

- *Strong convexity:* Unfortunately, our analysis does not benefit from strong convexity. However, using UAS of ERM for strongly convex losses + convergence rates for strongly convex SGD leads to UAS for SGD.

- *Motivating examples:* due to space considerations, we cannot include in the NeurIPS file detailed motivating examples of nonsmooth losses arising in ML. However, we will include those discussions in the full version. We also want to point out the reviewer is right in that when proximal mappings are efficiently computable, one should prefer using them: in particular, stability bounds for proximal methods are similar to those of smooth SGD, which is known since Hardt et al.'16. We do not include this discussion since it is a well known technique, and we address general settings when this operator may not be efficiently computable.

- *Gradients vs. subgradients:* We acknowledge the ambiguity in denoting subgradients by $\nabla f(x)$. However, since $\partial f(x)$ is a set (the subdifferential), not a vector, we prefer not to use $\partial$. We will add a comment clarifying that in the paper we denote by $\nabla f(x)$ any subgradient of $f$ at $x$.

- *About other works on stability:* We thank the reviewer for pointing out the "Fine-grained analysis" paper, which is a concurrent work to ours and appeared at ICML and on the arxiv both several weeks after the Neurips deadline. This work only addresses a weaker on-average stability notion, which in particular is not applicable to our privacy results. We will discuss this comparison in more detail in the revision. We also remark that approximate contraction may not hold for SGD in our setting. Indeed our lower bounds show that SGD trajectories can deviate $\Omega(\eta_t)$ at every step, which is the worst possible. Hence, our upper bounds are unimprovable, and they circumvent this requirement of approximate contractivity altogether.

  We were aware of "Private Stochastic Convex Optimization" paper, which also does not address uniform stability. More crucially, this paper has serious errors that impact all their main claims. We have contacted the authors of this paper, who have admitted these errors, and they will soon retract it from arXiv.

- *Privacy Analysis:* The privacy analysis follows from the prior work in almost a straightforward way. The technique in [1] applies to our case equally well. The only place where sampling is invoked in the analysis of [1] is in Lemma 3. In the proof of that lemma, it is easy to see that the only relevant condition that involves sampling is satisfied in our case. Note that in our algorithm we sample one point in each iteration, hence the distributions induced over a pair of neighboring datasets satisfy the same condition in the proof of Lemma 3 in [1] (where $q$ in that lemma is $1/n$). As for the asymptotic moment bound in [1], we can derive explicit bounds on the constants in [1] in our case when $n$ is sufficiently large. We will clarify all these details in the full version.

- *Optimality of SGD:* Unfortunately, we do not have space in the main file to add derivations regarding the optimal rates for multipass SGD. We will however explicitly mention this result, referring to the full version.

- *DP-SCO based on permutation SGD?* For fixed permutation SGD, we don't have privacy amplification, which we rely on in the construction based on sampling with replacement. For a random permutation (per epoch), we have some form of privacy amplification by sampling *without* replacement, but it is weaker than what we have in the sampling with replacement SGD. However, this approach could be interesting with some form of mini-batching (where batches are sampled without replacement).

[Meta-Review · NeurIPS 2020]

This paper got high scores: 9,7,6,8, all with high confidence. The major concerns are from Reviewer #3, who asked about the relationship with "Fine-Grained Analysis of Stability and Generalization for Stochastic Gradient Descent" ([*]) and whether the technique of using Poisson sampling in [1] can be used in the current work which uses uniform sampling. While the authors clarified in the rebuttal that their work is stronger than [*] and some of the results in [*] may not be generalized to the case in the current paper, and the technique in [1] can be applied straightforwardly, Reviewer #3 further refuted on the first claim and doubted on the second claim during discussion. The AC confirmed that this paper is concurrent with [*] and deemed that Reviewer #3 may have missed the sketched proof in the "Privacy Analysis" section of the rebuttal. During further discussion, Reviewer #3 acknowledged that the sketched proof made sense and supported acceptance. Reviewer #1 echoed. So the AC recommended acceptance.